# Importin-9 wraps around the H2A-H2B core to act as nuclear importer and histone chaperone

Abhilash Padavannil[1], Prithwijit Sarkar[2], Seung Joong Kim[3], Tolga Cagatay[1], Jenny Jiou[1], Chad A Brautigam[4], Diana R Tomchick[4], Andrej Sali[5,6], Sheena D'Arcy[7], Yuh Min Chook[1]*

[1]Department of Pharmacology, University of Texas Southwestern Medical Center, Dallas, United States; [2]Department of Biological Sciences, University of Texas at Dallas, Richardson, United States; [3]Department of Physics, Korea Advanced Institute of Science and Technology (KAIST), Daejeon, Korea; [4]Department of Biophysics, University of Texas Southwestern Medical Center, Dallas, United States; [5]Department of Bioengineering and Therapeutic Sciences, California Institute for Quantitative Biosciences, University of California, San Francisco, San Francisco, United States; [6]Department of Pharmaceutical Chemistry, California Institute for Quantitative Biosciences, University of California, San Francisco, San Francisco, United states; [7]Department of Chemistry and Biochemistry, University of Texas at Dallas, Richardson, United States

*For correspondence:
yuhmin.chook@utsouthwestern.edu

Competing interests: The authors declare that no competing interests exist.

**Abstract** We report the crystal structure of nuclear import receptor Importin-9 bound to its cargo, the histones H2A-H2B. Importin-9 wraps around the core, globular region of H2A-H2B to form an extensive interface. The nature of this interface coupled with quantitative analysis of deletion mutants of H2A-H2B suggests that the NLS-like sequences in the H2A-H2B tails play a minor role in import. Importin-9•H2A-H2B is reminiscent of interactions between histones and histone chaperones in that it precludes H2A-H2B interactions with DNA and H3-H4 as seen in the nucleosome. Like many histone chaperones, which prevent inappropriate non-nucleosomal interactions, Importin-9 also sequesters H2A-H2B from DNA. Importin-9 appears to act as a storage chaperone for H2A-H2B while escorting it to the nucleus. Surprisingly, RanGTP does not dissociate Importin-9•H2A-H2B but assembles into a RanGTP•Importin-9•H2A-H2B complex. The presence of Ran in the complex, however, modulates Imp9-H2A-H2B interactions to facilitate its dissociation by DNA and assembly into a nucleosome.
DOI: https://doi.org/10.7554/eLife.43630.001

## Introduction

Eukaryotic chromatin is organized into nucleosomes, which are structural and functional units that are composed of 147 base pairs of DNA wrapped around two H3-H4 dimers and two H2A-H2B dimers (*Luger et al., 1997*). Nucleosomes are assembled in the nucleus during S-phase as new H2A, H2B, H3 and H4 proteins are synthesized in the cytoplasm (*Adams and Kamakaka, 1999*; *Annunziato, 2012*; *Verreault, 2000*). Newly translated histones are folded and assembled into H2A-H2B and H3-H4 dimers, which are then imported into the nucleus for deposition onto replicating chromatin. Despite their small sizes, histones do not diffuse into the nucleus but are transported by nuclear import receptors of the Karyopherin-β family termed importins (*Baake et al., 2001*; *Jäkel et al.,*

**eLife digest** Cells contain two meters of DNA which, if left to its own devices, would soon end up in a knot. To keep things organized, the genetic code is wrapped around protein 'spools' called histones, meaning it can all fit within a part of the cell known as the nucleus. The cell makes a copy of its DNA every time it divides, and this copy needs a new set of histones to keep it tidy. The machinery required to construct new histones sits outside the nucleus and getting the histones into position in the nucleus can be a challenge. Histones have a positive charge, which helps to keep the negatively charged DNA wound around the spool. Yet without supervision, histones can stick to other charged molecules in the cell and cause blockages.

The proteins responsible for histone transport are called importins. These proteins normally recognize their cargo by molecular patterns called "nuclear localization signals". These patterns work like a postal address, telling the importin to take the cargo into the nucleus. When they arrive at their destination, another protein called Ran interacts with the importins to release the cargo. Strangely, removing the predicted address pattern from histones does not stop them getting to the nucleus. To find out what was going on, Padavannil et al. solved the three-dimensional structure of an importin bound to a pair of histones via a technique called X-ray crystallography. This made it possible to see how the proteins fit together.

The structure revealed that, rather than interact with the predicted address pattern, the importin wraps around the core of the histones. This blocks the positive charges, stopping the histones sticking to other molecules on their way to the nucleus. The next challenge was to find out how the cell unhooks the histone cargo from the importin when it arrives in the nucleus; with the positive charges covered by the importin, the histones could not stick to the DNA. Yet, something changed when the levels of Ran were high. Rather than unhook the histone, Ran joined the importin-histone complex. This then made it possible for the histones to attach to DNA, helping them to get into position without sticking to the wrong molecules.

These findings form the first step in understanding how the cell transports sticky histones without getting in a knot. The next step is to find out whether these interactions, shown in test tubes, happen in the same way inside living cells.

DOI: https://doi.org/10.7554/eLife.43630.002

---

1999; *Johnson-Saliba et al., 2000*; *Mosammaparast et al., 2002b*; *Mosammaparast et al., 2001*; *Mühlhäusser et al., 2001*).

Importins usually recognize their protein cargos by binding nuclear localization signals (NLSs) in their polypeptide chains. Importins bind nucleoporins to traverse the permeability barrier of the nuclear pore complex (NPC) (reviewed in *Chook and Süel, 2011*; *Cook et al., 2007*; *Görlich and Kutay, 1999*; *Izaurralde et al., 1997*; *Kim et al., 2018*; *Kosinski et al., 2016*; *Lin et al., 2016*; *Soniat and Chook, 2015*). The small GTPase Ran controls direction of transport. Binding of cargos and RanGTP to importins is mutually exclusive. In the nucleus, where Ran is kept in the GTP state by guanine nucleotide exchange factor RCC1, importins bind RanGTP with high affinity, resulting in cargo release (*Chook and Süel, 2011*; *Izaurralde et al., 1997*; *Soniat and Chook, 2015*).

Studies in importin-deletion yeast strains identified Kap114 (*S. cerevisiae* homolog of Importin-9 or Imp9) as the primary H2A-H2B importer, and Kap121 (homolog of Importin-5) and Kap123 (homolog, Importin-4) as secondary importers (*Mosammaparast et al., 2002b*; *Mosammaparast et al., 2001*). Pull-down binding from cytosolic HeLa extract and proteomics tracking nuclear-cytoplasmic localization in human cells also identified core histones as Imp9 cargos (*Jäkel et al., 2002a*; *Kimura et al., 2017*). The use of multiple backup importin systems is also seen in human cells, as many previous studies have shown that H2A and H2B can bind and be imported into nuclei of digitonin-permeabilized cells by several human importins (such as Importin-β, Karyopherin-β2, Importin-4, Importin-5, Importin-7) in addition to Importin-9 (*Baake et al., 2001*; *Johnson-Saliba et al., 2000*; *Mosammaparast et al., 2002b*; *Mosammaparast et al., 2001*; *Mühlhäusser et al., 2001*).

Core histones H2A, H2B, H3 and H4 all contain disordered N-terminal tails followed by small histone-fold domains; H2A also has a disordered C-terminal tail (*Luger et al., 1997*). The N-terminal tails of histones contain many basic residues, somewhat resembling classical NLS motifs

(*Blackwell et al., 2007*; *Ejlassi-Lassallette et al., 2011*; *Johnson-Saliba et al., 2000*; *Marchetti et al., 2000*; *Greiner et al., 2004*; *Mosammaparast et al., 2001*; *Moreland et al., 1987*). H2A and H2B tails are able to target heterologous proteins into the nucleus (*Mosammaparast et al., 2001*), but removal of the tails does not abolish localization of H2A-H2B in the nucleus (*Thiriet and Hayes, 2001*). Furthermore, analysis of seven different importins binding to H3 and H4 tails *vs.* full-length H3-H4 *vs.* H3-H4•Asf1 chaperone complex suggested that specificities for importin-binding reside not only in the tail 'NLSs' but also in the histone folds and the bound chaperone (*Soniat et al., 2016*).

Here, we solved the crystal structure of Imp9 bound to the full-length H2A-H2B dimer to understand how histones are recognized for nuclear import. The superhelical Imp9 wraps around the histone dimer. Most of the N-terminal tails of both H2A and H2B are disordered, and only five residues of the H2B tail contact Imp9. Binding of Imp9 blocks DNA and H3-H4 sites on H2A-H2B, and Imp9 prevents H2A-H2B from aggregating on DNA, consistent with a histone chaperone-like activity for Imp9. Unlike other importin-cargo complexes, RanGTP does not dissociate Imp9•H2A-H2B but binds the complex and enhances its dissociation by DNA. The Ran•Imp9•H2A-H2B complex is also able to promote H2A-H2B assembly into nucleosomes. Formation of the Ran•Imp9•H2A-H2B complex appears to modulate importin-histone interactions to facilitate histone deposition to nuclear targets such as the assembling nucleosome.

## Results

### Structure of the Imp9•H2A-H2B complex

The major nuclear importer for H2A-H2B in *S. cerevisiae* is Kap114 (*Mosammaparast et al., 2002b*; *Mosammaparast et al., 2001*). Imp9, the human homolog of Kap114, was previously shown to bind and import H2A-H2B (*Jäkel et al., 2002a*; *Kimura et al., 2017*; *Mühlhäusser et al., 2001*). We show Imp9-histone interactions in immunoprecipitation from the cytoplasmic fraction of a stable HeLa cell line expressing mCherry-H2B (*Figure 1A*). We also show by fluorescence microscopy that Imp9 in these cells localizes mostly to the cytoplasm (*Figure 1B*). Similar cytoplasmic localization of Imp9 was reported in the Human Protein Atlas (*Thul et al., 2017*; *Uhlen et al., 2017*). To understand how Imp9 recognizes histones for nuclear import, we solved the crystal structure of human Imp9 bound to full-length *X. laevis* H2A-H2B (dissociation constant, $K_D$ = 30 nM; *Table 1* and *Figure 1—figure supplement 1*) by single wavelength anomalous dispersion to 2.7 Å resolution (*Figure 1—source data 1*).

Imp9 is made up of twenty tandem HEAT repeats, each containing two antiparallel helices A and B that line the convex and concave surfaces of superhelical-shaped protein, respectively (*Figure 1C* and *Figure 1—figure supplement 2A,B*). The concave surface of Imp9 is mostly acidic, with a few small basic patches (*Figure 1—figure supplement 2B*). This charged concave surface of Imp9 wraps around H2A-H2B, burying 1352 Å$^2$ (26% of the H2A-H2B surface) at three distinct interfaces 1–3 (*Figure 2A–D* and *Figure 2—figure supplements 1–3*). The Imp9-bound H2A-H2B has a canonical histone-fold as in nucleosomes (151 Cα atoms aligned, r.m.s.d. 0.505 Å; PDB ID 1AOI) (*Luger et al., 1997*). In our structure, the N-terminal and C-terminal tails of H2A (residues 1–16, 101–130) and H2B (1-27, 125-126), the first 14 residues of Imp9 and its H19loop (residues 936–996) were not modeled due to missing electron density.

The N- and C-terminal HEAT repeats of Imp9 (Interfaces 1 and 3) clamp the histone-fold domain while the inner surface of central HEAT repeats 7–8 (Interface 2) interacts with a five-residue segment of the H2B N-terminal tail (*Figure 2*). Interface 1 on Imp9 comprises the loop that follows helix 2B and the last turns of helices 3B, 4B and 5B (*Figure 2B*, *Figure 2—figure supplements 1A*, *2A* and *3A*). Hydrogen-bonding with H2A-H2B residues caps the C-terminal ends of these Imp9 B helices (*Figure 2—figure supplement 1D*). Of note is the end-to-end capping of the last turn of Imp9 helix 4B by the first turn of histone H2B helix α2. Interface 1 on the histones involves α2-L2-α3 of H2A and α1-L1-α2 of H2B, which constitute a significant portion of the basic DNA-binding surface found in nucleosomes. Although histones and Imp9 surfaces at this interface are electrostatically complementary (*Figure 1—figure supplement 2B*), interactions also involve many hydrogen bonds, hydrophobic interactions and main chain interactions (*Figure 2—figure supplement 1A,D*).

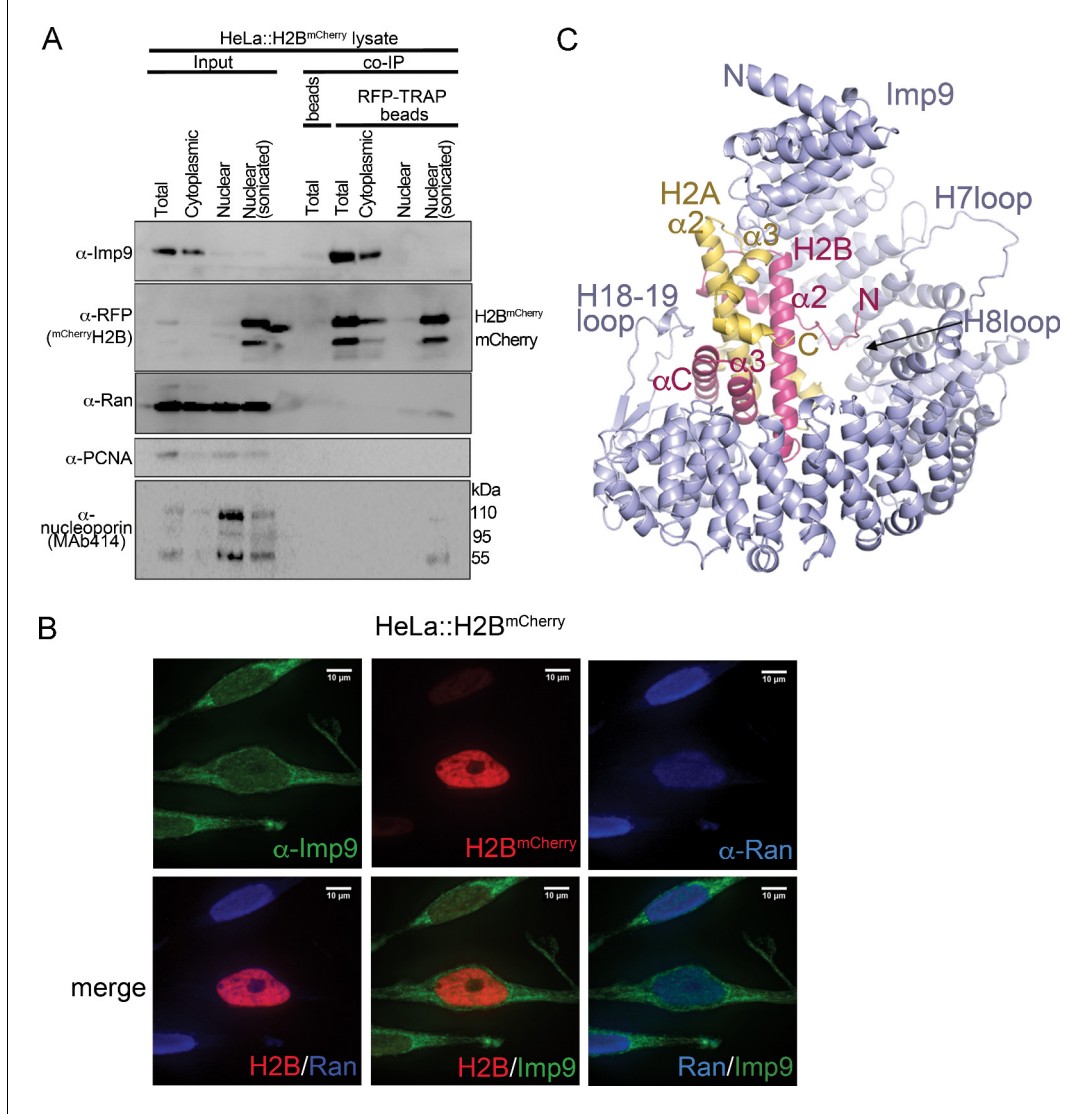

**Figure 1.** Interactions between Imp9 and H2A-H2B in the cell and crystal structure of the Imp9•H2A-H2B complex. (A) Coimmunoprecipitation (CoIP) studies of H2B^mCherry from whole cell, cytoplasmic and nuclear fractions of the lysates from HeLa cells stably expressing H2B^mCherry, followed by immunoblots with Imp9, Ran, RFP antibodies. PCNA and MAb414 antibodies are used as loading control antibodies. 10 μg of 1.5 mg lysates are analyzed as CoIP input. Blots are representative of three identical experiments. (B) Subcellular localization of Imp9 and Ran in Hela::H2B^mCherry cells. HeLa cells were fixed, permeabilized, incubated with affinity-purified rabbit polyclonal Imp9 antibody and mouse monoclonal anti–Ran antibody, and visualized by confocal microscopy. The secondary antibodies were Alexa 488 conjugated anti–rabbit and Alexa 405 conjugated anti-mouse, respectively. The column on the right contains two-color merge images. (C). The crystal structure of human Imp9 (blue) in complex with *X. laevis* H2A (yellow)-H2B (red).

DOI: https://doi.org/10.7554/eLife.43630.003

The following source data and figure supplements are available for figure 1:

**Source data 1.** Data collection and refinement statistics, Imp9•H2A-H2B structure.
DOI: https://doi.org/10.7554/eLife.43630.006
**Figure supplement 1.** ITC analysis of Imp9 binding to H2A-H2B.
DOI: https://doi.org/10.7554/eLife.43630.004
**Figure supplement 2.** HEAT repeat organization of Imp9 and electrostatic surface potential of Imp9 and H2A-H2B.
DOI: https://doi.org/10.7554/eLife.43630.005

**Table 1.** Imp9-H2A-H2B binding affinities by Isothermal Titration Calorimetry.

| Binding species | $K_D$(nM)[*] | ΔH (kCal/mol) | ΔS (Cal/mol.K) | ΔG (kCal/mol) | Imp9 concentration correction factor |
|---|---|---|---|---|---|
| Imp9 + H2A-H2B | 30 [10, 70][†] | −10.2 [−10.6, -9.8][‡] | −0.6 | −10.0 | 0.90 [0.88, 0.92][§] 0.90 [0.87, 0.93] 0.90 [0.88, 0.92] |
| Imp9 + H2AΔTail[¶] -H2B | 40 [20, 60] | −11.9 [−12.4,−11.5] | −6.7 | −10.0 | 0.83 [0.81,0.84] 0.86 [0.84,0.88] 0.85 [0.83,0.86] |
| Imp9 + H2A-H2BΔ(1-35) | 40 [10, 110] | −12.5 [−13.2,−11.9] | −8.5 | −10.0 | 0.87 [0.82, 0.91] 0.89 [0.86, 0.91] 0.87 [0.83, 0.91] |
| Imp9 + H2AΔTail-H2BΔTail[††] | 40 [10, 100] | −11.7 [−12.2,−11.2] | −5.9 | −9.9 | 1.0 [0.98, 1.03] 0.97 [0.92, 1.01] 0.92 [0.88, 0.96] |
| Imp9ΔH8loop + H2A-H2B | 10 [1, 20] | −10.1 [−10.4,−9.9] | 2.4 | −10.8 | 0.97 [0.96, 0.99] 1.06 [1.05, 1.07] 0.99 [0.98, 1.00] |
| Imp9ΔH18-H19loop + H2A-H2B | 450 [350, 600] | 7.9 [7.6, 8.3] | 56 | −8.5 | 1.12 [1.1, 1.2] 1.16 [1.12, 1.2] 1.15 [1.11, 1.19] |
| Imp9ΔH19loop + H2A-H2B | 40 [10, 100] | −11.0 [−11.4,−10.5] | −3.5 | −9.9 | 0.99 [0.98,1.02] 1.00 [0.98,1.03] 1.00 [0.97,1.04] |

[*] The $K_D$ value corresponds to a best-fit value obtained from global analysis of each experimental set carried out in triplicate.

[†] The 68.3% confidence interval for $K_D$ determined by global fit analysis of the triplicates in each experimental set.

[‡] The 68.3% confidence interval for ΔH determined by global fit analysis of the triplicates in each experimental set.

[§] The 68.3% confidence interval for concentration correction factor of Imp9 is determined by local fit analysis of each individual experiment in an experimental set of triplicates.

[¶] H2AΔTail – globular domain of H2A (residues 14–119).

[††] H2AΔTail-H2BΔTail - heterodimer of residues 14–119 of H2A with residues 25–123 of H2B.

The following supplement is available for **Table 1**:**Figure 1—figure supplement 1**

DOI: https://doi.org/10.7554/eLife.43630.007

Interface 2 involves Imp9 helices 7B, 8B and the H8loop (connects helices 8A to 8B) binding to the short [28]KKRRK[32] segment of the H2B N-terminal tail (**Figure 2C** and **Figure 2—figure supplements 1B**, **2B–C and** and **3B**). Electron densities for H2B [28]KKRRK[32] are weak (see **Figure 2—figure supplement 2B–C**) and atomic displacement parameters ('B-factors') for H2B residues 28–32 are also high (>100 Å[2]), suggesting dynamic interactions. Charged H2B side chains make electrostatic interactions with several acidic Imp9 residues, while the aliphatic part of these basic side chains and their backbone participate in hydrophobic interactions.

Interface 3 involves the last three HEAT repeats of Imp9, specifically the last turn of helix 18A and the short loop that follows, the H18-19loop, the C-terminal half of helix 19A and the first turn of helix 20B (**Figure 2D**, **Figure 2—figure supplements 1C**, **2D** and **3C**). Instead of the typical basic H2A-H2B residues interacting with the acidic Imp9 residues, charges at Interface 3 are reversed (**Figure 1—figure supplement 2B**). A basic patch formed by the Imp9 H18-19loop and nearby helices complement an acidic surface on the histones formed by residues from H2A helices α2 and αC, and the C-terminal half of H2B that comprises α2-α3-αC. Of note here are salt bridges between Imp9 residue Arg898 and several acidic residues of H2A (**Figure 2D**). Many hydrophobic contacts are also found at this interface, and several helices (Imp9 H18A, H19A and histone H2B α2) are capped through hydrogen-bonding with partner proteins (**Figure 2—figure supplements 1C** and **3C**).

## Distribution of binding energy in the Imp9•H2A-H2B complex

We analyzed the distribution of binding energy of the extensive Imp9-H2A-H2B interface through mutagenesis of the N-terminal histone tails and several long Imp9 loops and determined $K_D$s of the mutants using isothermal titration calorimetry (ITC; **Table 1** and **Figure 1—figure supplement 1**). Imp9 binds full-length H2A-H2B with high affinity ($K_D$ = 30 nM). We did not make mutations to

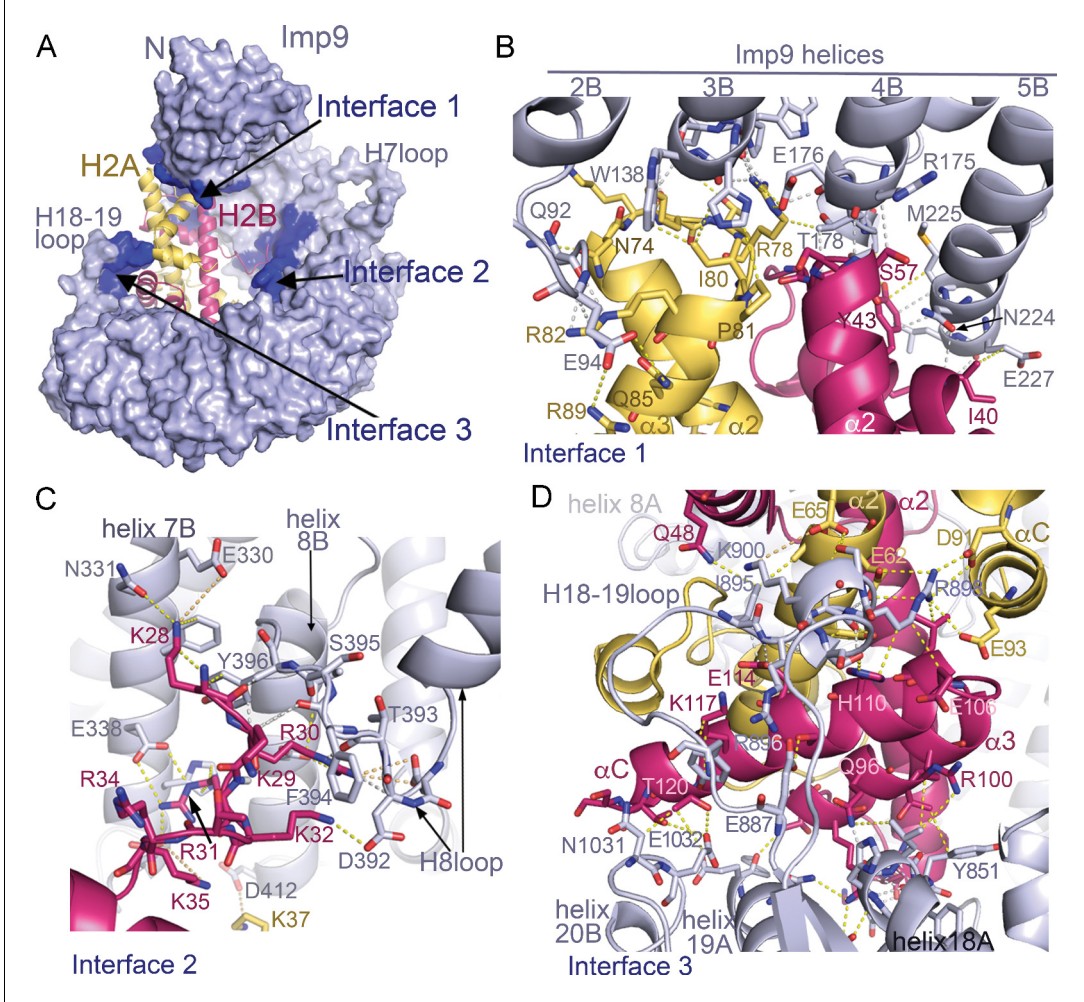

**Figure 2.** Imp9 •H2A-H2B binding interfaces. (**A**) The Imp9•H2A-H2B structure is oriented as in *Figure 1C*. The histones H2A (yellow)-H2B (red) are drawn as cartoons. Imp9 (blue) is represented as surface showing three distinct H2A-H2B binding interfaces (dark blue). (**B–D**). Details of Interface 1 (**B**), Interface 2 (**C**) and Interface 3 (**D**). Intermolecular contacts are shown as dashed lines.

DOI: https://doi.org/10.7554/eLife.43630.008

The following figure supplements are available for figure 2:

**Figure supplement 1.** Stereo views of the Imp9•H2A-H2B interfaces.

DOI: https://doi.org/10.7554/eLife.43630.009

**Figure supplement 2.** Stereo views of representative electron-density within the Imp9•H2A-H2B interfaces.

DOI: https://doi.org/10.7554/eLife.43630.010

**Figure supplement 3.** Sequence alignment of regions of Imp9 that interact with H2A-H2B.

DOI: https://doi.org/10.7554/eLife.43630.011

Interface 1 because of the many main-chain interactions found there (*Figure 2—figure supplement 1D*). Interface 2 involves the H8 loop of Imp9 and the N-terminal tail of H2B, both of which are convenient for deletion mutagenesis. Similarly, two long Imp9 loops (H18-19loop and H19loop) in Interface 3 are convenient for deletion mutagenesis.

H2A-H2B mutant assembled with the core of H2A (residues 14–119) and full-length H2B, hence named H2AΔTail-H2B, has similar binding affinity ($K_D$ = 40 nM) as full-length H2A-H2B. This result is consistent with structural observations that H2A residues in its N- and C-terminal tails are disordered and likely do not contact Imp9. Removal of the H2B tail (deleting residues 1–35), generating mutant H2A-H2BΔ(1-35), also did not affect binding affinity ($K_D$ = 40 nM). This is not surprising given the weak electron density and high B-factors of H2B [28]KKRRK[32] bound to Imp9 in Interface 2 (*Figure 2—*

*figure supplement 2*). A H2A-H2B mutant dimer with only the core domain (H2A residues 14–119 complexed with H2B residues 25–123; named H2AΔTail-H2BΔTail) also bind as tightly to Imp9 as the full-length histones ($K_D$ = 40 nM). Removal of the Imp9 H8loop (Imp9ΔH8loop), which forms part of the binding site for H2B $^{28}$KKRRK$^{32}$, also did not decrease binding ($K_{D,Imp9ΔH8loop}$ = 10 nM; *Table 1*, *Figure 1—figure supplement 1E*). The histone tails thus do not contribute much binding energy for interactions with Imp9.

At Interface 3, the basic H18-19loop of Imp9 contacts the acidic patch of the histones while the nearby H19loop is mostly disordered and its contribution to histone binding is uncertain. Removal of the H18-19loop reduced the affinity 15-fold ($K_D$ = 450 nM; *Table 1*, *Figure 1—figure supplement 1F*). We note the endothermic binding reaction that occurred upon truncation of this 40-residue loop. This result suggests substantial contribution of Interface three to the total binding energy. Removal of the H19loop did not affect affinity ($K_D$ = 40 nM; *Table 1*, *Figure 1—figure supplement 1G*), suggesting that this disordered loop does not participate in H2A-H2B binding.

## Imp9 functions biochemically like a histone chaperone

A large portion of the Imp9 interface on H2A-H2B overlaps with the DNA-binding and H3-H4-binding interfaces used in nucleosomes (*Figure 3A–C*). This feature of Imp9 occluding interfaces used in the nucleosome is common to many H2A-H2B histone chaperones of H2A-H2B (*Hammond et al., 2017*). Imp9 in fact buries more surface area on H2A-H2B (1352 Å$^2$) than well-characterized H2A-H2B chaperones such as Nap1 (387 Å$^2$), Swr1 (488 Å$^2$), Anp32e (533 Å$^2$), Chz1 (906 Å$^2$), Spt16 of FACT (185 Å$^2$) and YL1 (883 Å$^2$) (*Hondele et al., 2013*; *Hong et al., 2014*; *Kemble et al., 2015*; *Luger et al., 1997*; *Mosammaparast et al., 2002a*; *Obri et al., 2014*; *Zhou et al., 2008*).

Histone chaperones are a class of functionally, structurally and mechanistically diverse histone-binding proteins that 'chaperone' histones to protect them from promiscuous DNA-histone interactions (*Elsässer and D'Arcy, 2012*; *Mattiroli et al., 2015*) in many different contexts surrounding the formation of nucleosomes (*Laskey et al., 1978*). The observation that Imp9 buries more surface area on H2A-H2B than well-characterized histone chaperones raises the question of whether Imp9 might also function as a histone chaperone. This function is manifested biochemically by the protein out-competing DNA from non-nucleosomal DNA•H2A-H2B complexes (*Andrews et al., 2010*; *Andrews et al., 2008*; *Hondele et al., 2013*; *Hong et al., 2014*). To test if Imp9 can compete H2A-H2B from DNA like histone chaperone Nap1, we performed native gel-based competition assays. Titration of Nap1 or Imp9 against DNA•H2A-H2B complexes leads to the release of free DNA as Nap1 or Imp9 binds H2A-H2B (*Figure 3D,E*). These results suggest that Imp9 can act as a histone chaperone by shielding H2A-H2B from promiscuous interactions while it accompanies the histones from the cytoplasm to the nucleus.

## RanGTP does not release H2A-H2B but assembles to form RanGTP•Imp9•H2A-H2B

RanGTP generally binds importins with high affinity to dissociate Importin-cargo complexes and release cargos into the nucleus. However, this appears not to be the case with Imp9•H2A-H2B. When increasing concentrations of RanGTP (5–30 molar equivalents *S. cerevisiae* Ran(1–177/Q71L)) are added to an immobilized MBP-Imp9•H2A-H2B complex, the histones are not released (*Figure 4A*; controls shown in *Figure 4—figure supplement 1A,C*). The RanGTP protein used in these experiments is fully active as it easily dissociates a cargo/NLS from Kapβ2 (*Figure 4B* and *Figure 4—figure supplement 1B*). In a separate experiment, the Imp9•H2A-H2B complex also remains intact when added to immobilized MBP-RanGTP (*Figure 4C*). MBP-RanGTP binds to H2A-H2B-bound Imp9 to form what seems to be a heterotetrameric MBP-RanGTP•Imp9•H2A-H2B complex (*Figure 4C*).

We examined the interactions of Imp9•H2A-H2B with RanGTP in solution using electrophoretic mobility shift assays and size exclusion chromatography. Electrophoretic mobility shift assays (EMSA) show the formation of a 1:1 complex between Imp9 and H2A-H2B (*Figure 4D*) as well as between Imp9 and RanGTP (*Figure 4E*, lanes 3–6). A complex containing equimolar amounts of Imp9, H2A-H2B and RanGTP can also form (*Figure 4E*, lanes 7–10). Size exclusion chromatography of Imp9•H2A-H2B in the presence of excess RanGTP also shows a large complex that contains Imp9, H2A-H2B and Ran (*Figure 4—figure supplement 2*).

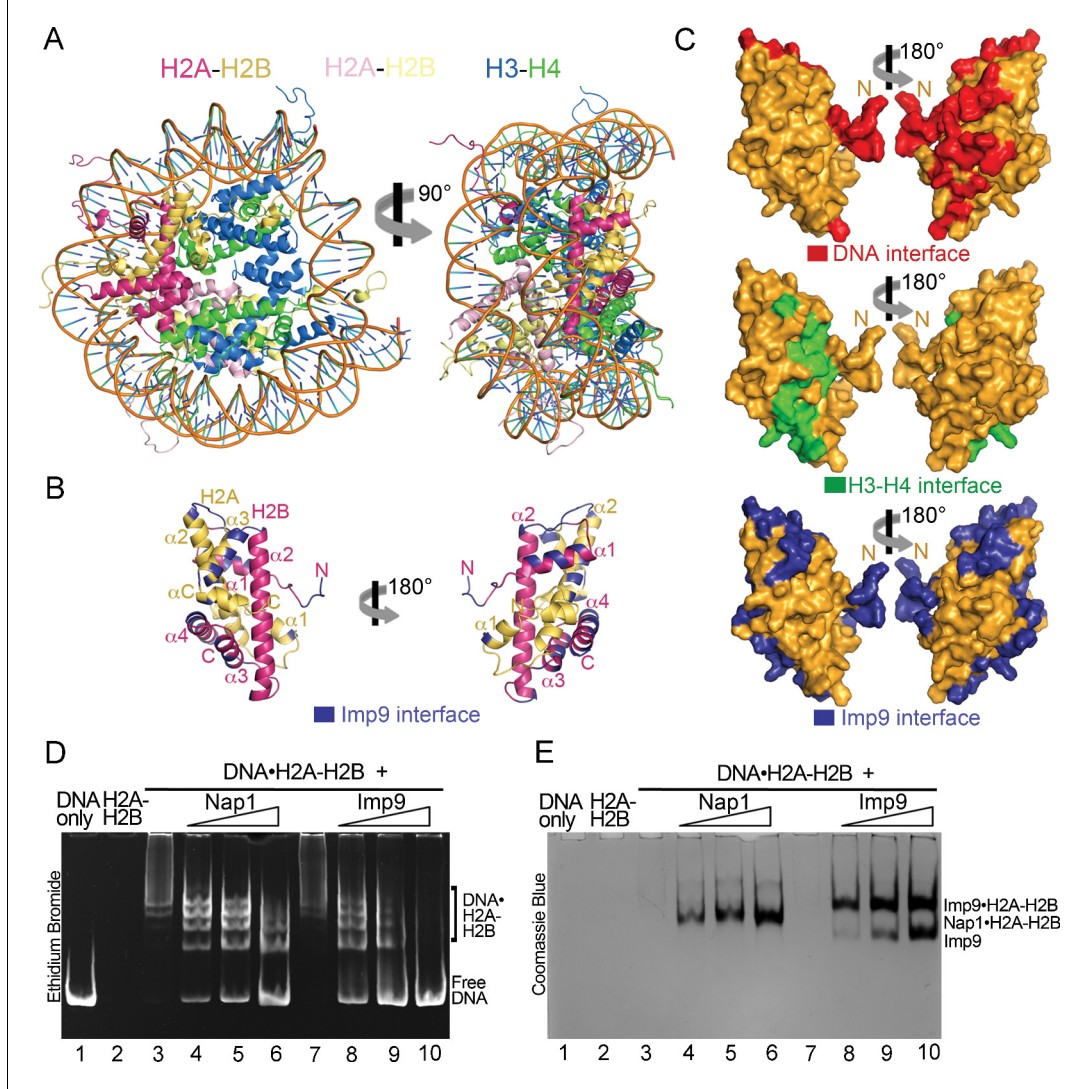

**Figure 3.** Imp9 has structural and biochemical characteristics of a histone chaperone. (A) Structure of the nucleosome (1AOI): the orientation on the right shows one of the H2A-H2B dimers (in red and yellow) in the same orientation as H2A-H2B shown in the right panel of B. (B) Imp9-bound H2A-H2B (Imp9 not shown) with its Imp9 interface in dark blue. Orientation of H2A-H2B on the left is the same as in *Figures 1C* and *2A*. (C) Surface representations of the H2A-H2B dimer surface (same orientation as in B) showing nucleosomal DNA (red), nucleosomal H3-H4 (green) and Imp9 (blue) binding interfaces. (D-E) Gel-shift assays to probe chaperone activity of Imp9. Increasing concentrations of Imp9 or Nap1 (0.5, 1.0 and 1.5 molar equivalents of H2A-H2B) were added to pre-formed DNA•H2A-H2B complexes, and the mixtures separated on a native gel stained with ethidium bromide to visualize DNA (D) and with Coomassie Blue to visualize protein (E). The two images of the same gel are horizontally aligned. The histone chaperone Nap1 binds H2A-H2B (E, lanes 4–6) leading to the release of free DNA (D, lanes 4–6). Imp9 also releases free DNA (D, lanes 8–10) as it binds H2A-H2B (E, lanes 8–10).

DOI: https://doi.org/10.7554/eLife.43630.012

We used analytical ultracentrifugation to rigorously and quantitatively assess the formation of a heterotetrameric RanGTP•Imp9•H2A-H2B complex. We examined individual Imp9, H2A-H2B and RanGTP proteins, equimolar mixes of Imp9+H2A-H2B and Imp9+RanGTP, and a 1:1:3 molar ratio mix of Imp9, H2A-H2B and RanGTP by analytical ultracentrifugation (protein concentrations 3–10 μM; *Figure 4F*). Sedimentation coefficient values of the individual proteins estimated from the sedimentation velocity experiments are consistent with their molecular weights: Imp9 (3.7S), H2A-H2B (1.3S) and RanGTP (1.4S). The binary complexes of Imp9•H2A-H2B and Imp9•RanGTP are both larger, at 4.3S and 4.2S, respectively. The mixture of Imp9, H2A-H2B and RanGTP gave peaks at

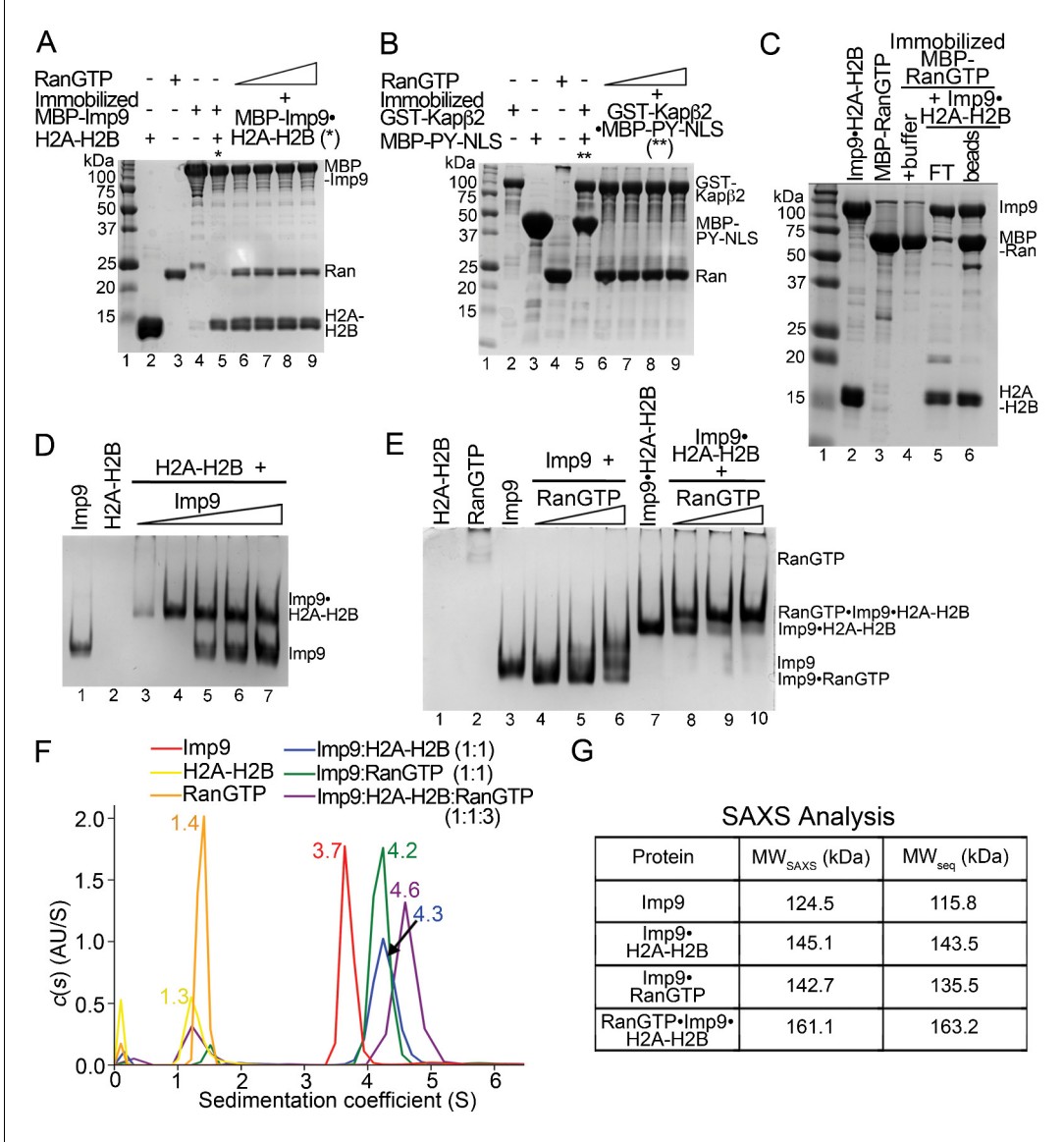

**Figure 4.** RanGTP does not release H2A-H2B but forms a RanGTP•Imp9•H2A-H2B complex. (**A**) Pull-down binding assay to probe RanGTP (*S. cerevisiae* Ran(1–179/Q71L)) interactions with the Imp9•H2A-H2B complex. Increasing concentrations of RanGTP (12.5 μM, 25 μM, 50 μM or 75 μM) were added to 2.5 μM MBP-Imp9•H2A-H2B that is immobilized on amylose resin. After extensive washing, the bound proteins were visualized by Coomassie-stained SDS-PAGE. Controls are shown in *Figure 4—figure supplement 1*. (**B**) Pull-down binding assays to show RanGTP mediated dissociation of the GST-Kapβ2•MBP-PY-NLS complex. Increasing concentrations of RanGTP (12.5 μM, 25 μM, 50 μM or 75 μM) were added to 2.5 μM GST-Kapβ2•MBP-PY-NLS (immobilized). After extensive washing, bound proteins were visualized by Coomassie-stained SDS-PAGE. Controls are shown in *Figure 4—figure supplement 1*. (**C**) Pull-down binding assay where preformed Imp9•H2A-H2B was added to immobilized MBP-RanGTP. After washing, the bound proteins were visualized by Coomassie-stained SDS-PAGE. (**D**) EMSA of Imp9 titrated at 0.5–2.5 molar equivalents to constant H2A-H2B. Upward shift of the Imp9 band shows that Imp9 interacts with H2A-H2B. (**E**) EMSA of Ran titrated at 1–3 molar equivalents to constant Imp9 (lanes 3–6) or Imp9•H2A-H2B (lanes 7–10). Downward shift of the Imp9 band shows that Imp9 interacts with Ran to form Imp9•RanGTP (compare lanes 4–6 to lane 3), while upward shift of the Imp9•H2A-H2B band shows that a heterotetrameric Ran•Imp9•H2A-H2B complex forms (compare lanes 8–10 to lane 7). No Imp9 or Imp9•RanGTP band is present in lanes 8–10 indicating no dissociation of the Imp9•H2A-H2B complex by RanGTP. Proteins inputs for lanes 1–10 are shown in *Figure 4—figure supplement 1D*. (**F**) Analytical ultracentrifugation produced sedimentation profiles for Imp9, H2A-H2B, RanGTP, the 1:1 molar ratio mix of Imp9 and H2A-H2B dimer, the 1:1 molar ratio mix of Imp9 and RanGTP, and the 1:1:3 molar ratio mix of Imp9, H2A-H2B dimer and RanGTP. (**G**) Molecular weights estimated from merged SAXS profiles (MW$_{SAXS}$) for Imp9, Imp9•H2A-H2B, Imp9•RanGTP, and RanGTP•Imp9•H2A-H2B, compared with molecular weights from the protein sequences (MW$_{seq}$).

DOI: https://doi.org/10.7554/eLife.43630.013

The following source data and figure supplements are available for figure 4:

**Source data 1.** Summary of the SAXS experiments and analysis.

*Figure 4 continued on next page*

*Figure 4 continued*

DOI: https://doi.org/10.7554/eLife.43630.020

**Source data 2.** Summary of the SAXS parameters.

DOI: https://doi.org/10.7554/eLife.43630.021

**Figure supplement 1.** Interactions between RanGTP and the Imp9•H2A-H2B complex.

DOI: https://doi.org/10.7554/eLife.43630.014

**Figure supplement 2.** Size exclusion chromatography of Imp9, H2A-H2B and RanGTP complexes.

DOI: https://doi.org/10.7554/eLife.43630.015

**Figure supplement 3.** SAXS analysis of Imp9, Imp9•RanGTP, Imp9•H2A-H2B, and RanGTP•Imp9•H2A-H2B.

DOI: https://doi.org/10.7554/eLife.43630.016

**Figure supplement 4.** Comparative structural analysis of importin-RanGTP complexes.

DOI: https://doi.org/10.7554/eLife.43630.017

**Figure supplement 5.** RanGTP binding interfaces at HEAT repeats 1–4 of Kap95, Kap121, Importin-β, Importin-13 and Transportin-SR2.

DOI: https://doi.org/10.7554/eLife.43630.018

**Figure supplement 6.** The predicted RanGTP binding site at HEAT repeats 1–4 of Imp9.

DOI: https://doi.org/10.7554/eLife.43630.019

1.4S (excess RanGTP) and 4.6S. The 4.6S assembly is larger than either Imp9•H2A-H2B or Imp9•RanGTP and is likely the quaternary RanGTP•Imp9•H2A-H2B complex.

We also studied Imp9, Imp9•RanGTP, Imp9•H2A-H2B, and RanGTP•Imp9•H2A-H2B (protein concentrations 31–43 µM) by small angle X-ray scattering (SAXS). SAXS profiles for the four Imp9-containing samples were analyzed to calculate radius of gyration ($R_g$), maximum particle size ($D_{max}$) and pair distribution function (P(r (*Figure 4G*, *Figure 4—figure supplement 3*, *Figure 4—source datas 1* and *2*). The linearity of the Guinier plots confirms a high degree of homogeneity for each of the SAXS samples (*Figure 4—figure supplement 3A–D*). Molecular weight of the RanGTP•Imp9•H2A-H2B complex was estimated to be 161.1 KDa by using SAXS MOW (*Fischer et al., 2010*) a value nearly identical to the expected molecular weight of 163.2 kDa from the sequence thus confirming stability of the 4-polypeptide chain RanGTP•Imp9•H2A-H2B complex in solution (*Figure 4G*).

We compared the Imp9•H2A-H2B structure with the structures of different importins bound to RanGTP, to predict the Ran-binding site on Imp9. In these structures, RanGTP is always sandwiched between N-terminal and either central or C-terminal HEAT repeats of the importins (*Figure 4—figure supplement 4*). Importin-RanGTP interactions at the first four HEAT repeats of importins (binding Switch 1, Switch two and α3 of RanGTP) appear to be structurally conserved even though the interface on the opposite side of RanGTP involves different central or C-terminal HEAT repeats in different importins (*Chook and Blobel, 1999*; *Kobayashi and Matsuura, 2013*; *Lee et al., 2005*; *Tsirkone et al., 2014*; *Vetter et al., 1999*). Structural alignment of HEAT repeats 1–4 of Imp9 with HEAT repeats 1–4 of Importin-β(1-462)•RanGTP (PDB ID 1IBR (*Vetter et al., 1999*); r.m.s.d. of 152 Cαs in the alignment is 3.27 Å), Kap95•RanGTP (2BKU (*Lee et al., 2005*) r.m.s.d. of 152 Cαs in the alignment is 3.20 Å), Kapβ2•RanGTP (1QBK (*Chook and Blobel, 1999*) r.m.s.d. of 152 Cαs in the alignment is 4.02 Å), Kap121•RanGTP (3W3Z (*Kobayashi and Matsuura, 2013*); r.m.s.d. of 144 Cαs in the alignment is 2.51 Å), Transportin-SR2•RanGTP (4C0Q; (*Maertens et al., 2014*) r.m.s.d. of 144 Cαs in the alignment is 5.02 Å) and Importin-13•RanGTP (2 × 19 (*Bono et al., 2010*); r.m.s.d. of 144 Cαs in the alignment is 3.29 Å), and examination of the six structures at a single orientation of Imp9, show that RanGTP binds in very similar orientations to very similar locations at the N-terminus of these importins (*Figure 4—figure supplement 4A–F*). Examination of the interactions between the N-terminal HEAT repeats of Kap95, Kap121, Importin-β, Importin-13 and Transportin-SR2 with RanGTP, together with the sequence alignment of this region of the importins show positional/structural conservation of many interacting and potentially interacting (in Imp9) residues (*Figure 4—figure supplement 5A–G*). These analyses suggest that the N-terminal HEAT repeats of Imp9 are likely to be important in binding RanGTP.

Structural alignment of HEAT repeats 1–4 of Imp9 and Kap121•RanGTP allows us to predict the RanGTP binding site at the N-terminus of Imp9 (*Figure 4—figure supplement 6A,B*). The prediction is supported by an Imp9 mutant with HEAT repeats 1–3 removed that no longer binds RanGTP (*Figure 4—figure supplement 6C–E*). This likely Ran-binding site at the N-terminus of Imp9 appears separate from but adjacent to the H2A-H2B binding site (*Figure 4—figure supplement 6A,B*). The

GTPase can most likely access Imp9 without dislodging H2A-H2B but proximity of RanGTP to the histones could modulate Imp9-histones interactions especially the kinetics of binding.

## RanGTP•Imp9•H2A-H2B is tuned to release histones for nucleosome assembly

We performed native gel-based competition assays to titrate DNA against Imp9•H2A-H2B or RanGTP•Imp9•H2A-H2B. DNA is unable to compete H2A-H2B from Imp9•H2A-H2B (*Figure 3D–E* and *Figure 5A*, lanes 5–7) but can compete H2A-H2B from RanGTP•Imp9•H2A-H2B to produce Imp9•RanGTP and DNA•H2A-H2B (*Figure 5A–B*, lanes 8–10). Unlike Imp9, which efficiently displaces DNA from the DNA•H2A-H2B complex (*Figure 5C–D*, lanes 4–6), Imp9•RanGTP does not displace DNA from the DNA•H2A-H2B complex (*Figure 5C–D*, lanes 8–10). These results show that the interaction between Imp9 and H2A-H2B is altered by RanGTP.

We next tested the ability of Imp9 and Imp9•RanGTP to assemble and disassemble nucleosomes (*Figure 5E,F*). Like Nap1, Imp9 and Imp9•RanGTP do not influence the stability of the tetrasome (*Figure 5—figure supplement 1A*). To monitor nucleosome assembly, we titrated H2A-H2B alone or with Nap1, Imp9 or Imp9 +RanGTP against tetrasome and assayed the formation of nucleosomes (*Figure 5E*). Nucleosomes form from H2A-H2B alone or with H2A-H2B and Nap1 or Imp9 +RanGTP (*Figure 5E*, lanes 4–5 and 8–9) but not with Imp9 alone (*Figure 5E*, lanes 6–7). Imp9 will bind H2A-H2B preventing its deposition on tetrasomes to make a nucleosome (*Figure 5—figure supplement 1B*, lanes 6–7). Notably, in the presence of RanGTP, Imp9 is better at promoting H2A-H2B deposition than either Nap1 or no chaperone. To monitor nucleosome disassembly, we titrated Nap1, Imp9, or Imp9 +RanGTP against nucleosomes (*Figure 5F*). We see that Imp9 can extract H2A-H2B from the nucleosome to produce tetrasome and Imp9•H2A-H2B (*Figure 5F*, lanes 5–6; *Figure 5—figure supplement 1C*), while Nap1 and Imp9 +RanGTP have no effect (*Figure 5F*, lanes 3–4 and 7–8). These data reinforce the chaperone-like activity of Imp9 and show that Ran influences the interaction between Imp9 and H2A-H2B, possibly through an allosteric mechanism as comparative analysis with other importin•RanGTP complexes suggests that the RanGTP binding site does not overlap with the H2A-H2B binding site. The RanGTP binds the Imp9•H2A-H2B complex to modulate importin-histones interactions to facilitate release of the histones for nucleosome assembly.

## Discussion

The solenoid-shaped Imp9 wraps around the folded globular domain of the H2A-H2B dimer, leaving most of the N-terminal tails of H2A, H2B and the C-terminal tail of H2A disordered in the complex. Only the 5-residue $^{28}$KKRRK$^{32}$ segment of the H2B tail contacts Imp9 even though weak electron density and high atomic displacement parameters of the H2B tail suggests that these interactions are dynamic. Our structural observations that Imp9 binds mostly to the globular domain of the H2A-H2B are also consistent with the lack of effect in Imp9 binding when either or both histone tails are deleted (*Table 1*), and with the previously reported nuclear localization of a mutant of H2A-H2B that lacks both its N-terminal tails (*Thiriet and Hayes, 2001*). However, very weak dynamic/fuzzy long-range electrostatic interactions between Imp9 and histones tails may still exist - we had previously reported very weak and dynamic interactions between an importin-cargo pair by NMR that could not be observed by X-ray crystallography or detected in mutagenesis/ITC experiments (*Yoshizawa et al., 2018*). Nevertheless, H2A-H2B thus belongs to a small category of nuclear import cargos that mostly use surfaces of folded domains rather than extended linear nuclear import/localization motifs to bind their importins (*Aksu et al., 2016*; *Bono et al., 2010*; *Cook et al., 2009*; *Grünwald and Bono, 2011*; *Grünwald et al., 2013*; *Matsuura and Stewart, 2004*; *Okada et al., 2009*).

Imp9-binding blocks both the nucleosomal DNA- and H3-H4-binding sites of H2A-H2B in a manner that is reminiscent of histone chaperone-H2A-H2B interactions. Interestingly, the Imp9•H2A-H2B binding interface is much larger than any known complexes of H2A-H2B or H2A.Z/H2B bound to histone chaperones (*Elsässer and D'Arcy, 2012*; *Mattiroli et al., 2015*). Imp9 also acts biochemically like a histone chaperone to prevent H2A-H2B from aggregating with DNA in vitro. The ability of Imp9 to structurally sequester H2A-H2B from promiscuous interactions with DNA and its function in trafficking the histones fit with the broadly defined class of histone-binding proteins known as histone chaperones (*Elsässer and D'Arcy, 2012*; *Mattiroli et al., 2015*). It is also generally thought

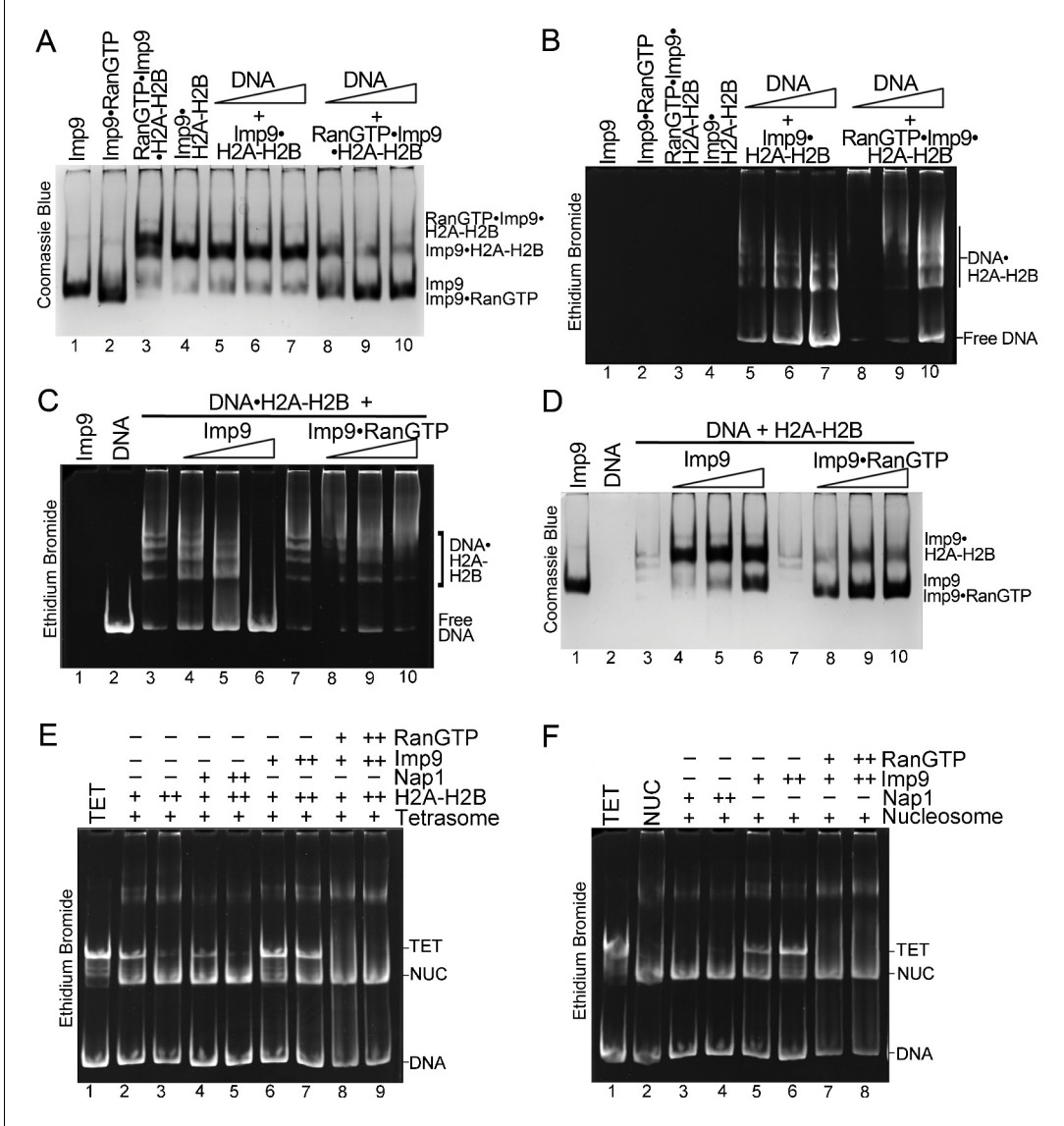

**Figure 5.** RanGTP modulates Imp9-H2A-H2B interactions for H2A-H2B deposition.  (A, B) DNA is titrated at 0.5, 1 and 2 molar equivalents of preformed Imp9•H2A-H2B (equimolar Imp9 and H2A-H2B mixed together) or RanGTP•Imp9•H2A-H2B (equimolar Imp9, H2A-H2B and RanGTP added together). Images of the same native gel, Coomassie stained in (A) and ethidium bromide stained in (B), are aligned for comparison. DNA cannot compete for H2A-H2B from the Imp9•H2A-H2B, leaving free DNA (B, increasing amounts from lanes 5 to 7) and intact Imp9•H2A-H2B (A, lanes 5–7). In contrast, DNA can compete for H2A-H2B from RanGTP•Imp9•H2A-H2B resulting in Imp9•RanGTP complexes (A, lanes 8–10), DNA•H2A-H2B complexes and very little free DNA (B, lanes 8–10). (C, D) Imp9 or Imp9•RanGTP (equimolar Imp9 and RanGTP added together) is titrated at 0.5–1.5 molar equivalents of H2A-H2B (in a DNA•H2A-H2B 1:7 complex). Images of the same native gel, ethidium bromide stained in (C) and Coomassie stained in (D), are aligned for comparison. Imp9 releases free DNA from DNA•H2A-H2B (C, lanes 3–6) and binds histones to form an Imp9•H2A-H2B complex (D, lanes 4–6). By comparison, Imp9•RanGTP releases little free DNA from DNA•H2A-H2B (C, lanes 7–10). (E) The presence of RanGTP and Imp9 facilitates H2A-H2B deposition onto the nucleosome. Nucleosome assembly assay where either H2A-H2B, Nap1•H2A-H2B, Imp9•H2A-H2B or RanGTP•Imp9•H2A-H2B is titrated in molar equivalents of 0.5 and 0.75 to tetrasome (TET; 2.5 µM). Nap1 and Imp9•RanGTP can form nucleosomes (NUC) while Imp9 cannot. Coomassie staining in *Figure 5—figure supplement 1B*. (F) Nucleosome disassembly assay where either Nap1, Imp9 or Imp9•Ran is titrated in molar equivalents of 0.5 and 0.75 to constant nucleosome (NUC; 2.5 µM). Imp9 can disassemble nucleosomes to tetrasomes while Nap1 and Imp9-Ran cannot. Coomassie staining in *Figure 5—figure supplement 1C*.

DOI: https://doi.org/10.7554/eLife.43630.022

The following figure supplement is available for figure 5:

**Figure supplement 1.** RanGTP modulates Imp9-histones interaction for H2A-H2B deposition.
DOI: https://doi.org/10.7554/eLife.43630.023

that there is little to no free histones in the cell as they are either bound in nucleosomes or by histone chaperones (*Elsässer and D'Arcy, 2012*; *Mattiroli et al., 2015*). Imp9 binds H2A-H2B in the cytoplasm, acts as a storage chaperone in the cytoplasm and a nuclear import receptor to take histones through the NPC (*Jäkel et al., 2002a*; *Kimura et al., 2017*; *Mosammaparast et al., 2002a*; *Mosammaparast et al., 2001*; *Mühlhäusser et al., 2001*).

Görlich and colleagues proposed in 2002 that negatively charged importins act as chaperones toward positively charged cargo proteins like histones (*Jäkel et al., 2002b*). Others also suggested importins acting as chaperones (*Lusk et al., 2002*). We provide structural evidence to support this proposal as the mostly negatively charged Imp9 indeed shields the mostly positively charged histone-fold domain of H2A-H2B, and perhaps also dynamically shields the extended basic histone tails. The ability of Imp9 to chaperone H2A-H2B, however, goes beyond charge shielding. Despite overall charge complementarity, there are only a few salt bridges at the Imp9•H2A-H2B interface, which also employs hydrophobic interactions and hydrogen bonds, many involving main chains of both proteins. Imp9 also shields many hydrophobic patches on H2A-H2B. The interaction further involves a charge reversal where a basic surface at the C-terminal end of Imp9 interacts with the acidic patch of H2A-H2B. The extensive and persistent interactions that allows Imp9 to surround and shield H2A-H2B also differ significantly from the recently revealed chaperoning interactions of another importin, that of Kapβ2 (or Transportin-1) with the Fused in Sarcoma protein (FUS). Kapβ2-FUS interactions are anchored through high affinity binding at the 26-residue PY-NLS linear motif of FUS that then enable weak, distributed and dynamic interactions with multiple mostly intrinsically disordered regions of FUS, to block formation of higher-order FUS assemblies and liquid-liquid phase separation (*Yoshizawa et al., 2018*).

The way the Imp9 solenoid wraps around H2A-H2B leaves the predicted N-terminal Ran-binding site of Imp9 accessible and ready to bind RanGTP. We showed by pull-down, electrophoretic mobility shift, size exclusion chromatography, analytical ultracentrifugation and SAXS experiments that Imp9 binds both the histones and RanGTP simultaneously and stably, suggesting that unlike most importin-cargo complexes, Imp9•H2A-H2B is unlikely to be dissociated by RanGTP alone upon entering the nucleus. This finding is not without precedence as Pemberton and colleagues previously showed an assembly that contains Kap114, H2A-H2B, RanGTP and the histone chaperone Nap1 (*Mosammaparast et al., 2002a*). Unlike the Pemberton study, which found the complex containing RanGTP, histones and importin to be intact in the yeast nucleus, we do not detect interactions between Imp9 and H2A-H2B in the nucleus even though that interaction is easily observed in the cytoplasm (*Figure 1A,B*). Imp9 is likely dissociated from histones soon after the complex enters the nucleus.

We showed that RanGTP changes the interactions between Imp9 and H2A-H2B as it forms the RanGTP•Imp9•H2A-H2B complex. DNA competes effectively with RanGTP•Imp9•H2A-H2B to produce Imp9•RanGTP and DNA•H2A-H2B even though it is unable to extract H2A-H2B from Imp9•H2A-H2B. Furthermore, RanGTP•Imp9•H2A-H2B is better at promoting H2A-H2B deposition to assemble nucleosome than either Nap1•H2A-H2B or no chaperone, while Imp9 alone cannot deposit H2A-H2B. The GTPase in the RanGTP•Imp9•H2A-H2B complex appears to modulate Imp9-H2A-H2B interactions to facilitate histone release and nucleosome assembly. Accessibility of the N-terminal HEAT repeats of Imp9 in the histones complex may allow formation of the RanGTP•Imp9•H2A-H2B complex, but proximity of the Ran and histones binding sites coupled with the flexibility of the HEAT repeats architecture of Imp9 and the propensity for conformational changes likely changed the kinetics of Imp9-histone binding.

Although histones can be deposited by RanGTP•Imp9•H2A-H2B onto DNA or the tetrasome, it remains unclear how H2A-H2B is released from Imp9 in cells. Assembling nucleosomes may release H2A-H2B from RanGTP•Imp9•H2A-H2B or the histones may be passed to another histone chaperone or nucleosome assembly factor as part of a chaperone hand-off cascade in the nucleus. These questions and the one regarding potential additional roles for Imp9 in the cytoplasm are topics for future studies.

## Materials and methods

### Constructs, protein expression and purification

Wild-type human Imp9 and Imp9 mutants (Imp9ΔH8loop, residues 371–396 replaced with SGSTGGSGS linker; Imp9ΔH18-19loop, residues 890–906 replaced with GSGTGSGSS; Imp9ΔH19loop, residues 941–996 (GGS)$_{12}$) were cloned into the pGEX-4T3 vector (GE Healthcare, USA) or the p*malE* vector (New England BioLabs, Ipswich, MA) modified to contain a TEV cleavage site (*Chook and Blobel, 1999*; *Chook et al., 2002*) and express His$_6$-MBP instead of MBP (pHis$_6$-Mal-TEV). Plasmids expressing the *X. laevis* histones H2A and H2B were a gift from Bing Li, UT Southwestern Medical Center. The construct for mutant H2BΔ(1-35) was PCR-amplified from the wildtype H2B construct and cloned into pET-3A vector (Novagen, USA).

Imp9 and Imp9 mutants were expressed in BL21 (DE3) *E.coli* cells (induced with 0.5 mM isopropyl-β-d-1-thiogalactoside (IPTG) for 12 hr at 20°C). Cells were harvested by centrifugation, resuspended in lysis buffer (50 mM Tris-HCl (pH 7.5), 0.1 mM NaCl, 1 mM EDTA, 2 mM DTT, 20% glycerol and complete protease inhibitors (Roche Applied Science, Mannheim, Germany)) and then lysed with the EmulsiFlex-C5 cell homogenizer (Avestin, Ottawa, Canada). GST-Imp9 was purified using Glutathione Sepharose 4B (GSH; GE Healthcare) and the GST tag was cleaved using Tev protease on the GSH column. Imp9 was further purified by anion exchange chromatography followed by size-exclusion chromatography (Superdex200, GE Healthcare; final buffer – 20 mM HEPES (pH 7.3), 110 mM potassium acetate, 2 mM magnesium acetate, 2 mM DTT, 15% glycerol). MBP-Imp9 was purified using amylose resin (NEB) affinity chromatography. The MBP tag was left intact on MBP fusion proteins, which were used for in vitro pull-down binding assays and analysis by size exclusion chromatography.

Wild type and mutant *Xenopus* histones H2A, H2B proteins were expressed individually in *E.coli* BL21 DE3 *plysS* cells, which were lysed by sonication. The lysate centrifuged at 16000 rpm and the washed pellet was resuspended in unfolding buffer (7 M guanidinium HCl, 20 mM Tris HCl, pH 7.5, 10 mM DTT) and dialyzed overnight in SAU-200 buffer (7 M urea, 20 mM sodium acetate, pH 5.2, 200 mM NaCl, 1 mM EDTA, 5 mM β-mercaptoethanol). The unfolded histone protein samples were further purified with cation exchange chromatography in SAU buffer (200–600 mM NaCl) followed by dialysis overnight in cold water. Mutant H2BΔ(1-35) was purified as described above. Mutant proteins H2AΔTail (contains residues 14–119 of H2A) and H2BΔTail (contains residues 25–123 of H2B) used for Isothermal titration calorimetry were obtained from The Histone Source (Colorado, United States).

H2A-H2B was reconstituted by mixing equimolar concentrations of H2A and H2B in unfolding buffer followed by overnight dialysis into refolding buffer (2 M NaCl, 10 mM Tris HCl, 1 mM EDTA, 5 mM β-mercaptoethanol). The dialyzed sample was concentrated and purified using size-exclusion chromatography in refolding buffer (*Luger et al., 1997*). Mutant histone dimers (H2A- H2AΔTail-H2B, H2A-H2BΔ(1-35) and H2AΔTail-H2BΔTail) were reconstituted and purified as described above for full-length wild type H2A-H2B (*Luger et al., 1997*).

His-tagged full-length *S. cerevisiae* Nap1(C200A/C249A/C272A) in pHAT4 vector was expressed in BL21 (DE3) *E. coli* cells. Nap1 was purified by affinity chromatography using a GE HisTrap SP FF column followed by ion-exchange chromatography using GE Mono Q 10/100 GL column and gel filtration chromatography using GE Superdex-200 16/600 column (20 mM Tris pH 7.5, 300 mM NaCl, 0.5 mM TCEP).

Ran (Gsp1 (1–179, Q71L)) and MBP-Ran were expressed in *E.coli* BL21 (DE3) cells as His$_6$ –tag proteins (induced with 0.5 mM IPTG for 12 hr at 20°C). Harvested cells were lysed with the EmulsiFlex-C5 cell homogenizer (Avestin, Ottawa, Canada) and the proteins purified by affinity chromatography on Ni-NTA column. Eluted proteins were loaded with GTP, and RanGTP and MBP-RanGTP were further purified by cation exchange chromatography followed by exchanging into buffer containing 20 mM HEPES (pH 7.5), 100 mM NaCl, 4 mM magnesium acetate, 1 mM DTT, 10% glycerol (*Chook and Blobel, 1999*; *Fung et al., 2015*).

## Imp9•H2A-H2B complex assembly, crystallization, crystal structure determination

Purified Imp9 was mixed with 10-fold molar excess of H2A-H2B in gel filtration buffer (20 mM HEPES (pH 7.3), 110 mM potassium acetate, 2 mM magnesium acetate, 2 mM DTT, 15% glycerol). Imp9•H2A-H2B was separated from excess histones by size-exclusion chromatography and concentrated to 18 mg/ml for crystallization. Selenomethionyl-labeled Imp9 was expressed as described previously (*Doublié, 1997*) and purified as for Imp9. Selenomethionyl-Imp9•H2A-H2B complex was assembled as for the native complex. Initial crystals were obtained by the sitting drop vapor diffusion method from commercial screens (reservoir solution - 40 mM MES pH 6.5, 3 M potassium formate, and 10% glycerol) and were further optimized by the hanging drop vapor diffusion method. Crystals were cryoprotected in reservoir solution that was supplemented with 15% glycerol, and flash frozen in liquid nitrogen. Selenomethionyl-Imp9•H2A-H2B crystals were obtained in the same conditions as native crystals and were prepared similarly for crystallographic data collection.

Imp9•H2A-H2B native crystals diffracted to a minimum Bragg spacing ($d_{min}$) of 2.70 Å and exhibited the symmetry of space group $P2_12_12$ with cell dimensions of a = 127.4 Å, b = 223.3 Å, c = 131.8 Å and contained two heterotrimers per asymmetric unit. All diffraction data were collected at beamline 19-ID (SBC-CAT) at the Advanced Photon Source (Argonne National Laboratory, Argonne, Illinois, USA) and processed in the program *HKL-3000* (*Minor et al., 2006*) with applied corrections for effects resulting from absorption in a crystal and for radiation damage (*Borek et al., 2003*; *Otwinowski et al., 2003*), the calculation of an optimal error model, and corrections to compensate the phasing signal for a radiation-induced increase of non-isomorphism within the crystal (*Borek et al., 2010*; *Borek et al., 2013*). These corrections were crucial for successful phasing and stable model refinement. Crystals of Imp9•H2A-H2B displayed mildly anisotropic diffraction. To minimize radiation damage and maximize redundancy, native data was collected in two separate scans of 125 degrees for a total of 250 degrees by translating a single crystal in the X-ray beam. Analysis of the self-Patterson function calculated with the native data revealed a significant off-origin peak at approximately (1/2, 1/2, 1/2) and 27% the height of the origin peak, indicating translational pseudosymmetry.

Phases were obtained from a single wavelength anomalous dispersion (SAD) experiment using the selenomethionyl-Imp9•H2A-H2B protein with data to 2.65 Å. Fifty-four selenium sites were located, phases improved and an initial model containing over 50% of all Imp9•H2A-H2B residues was automatically generated in the *AutoBuild* routine of the *Phenix* (*Adams et al., 2010*) program suite. Completion of this model was performed by manual rebuilding in the program *Coot* (*Emsley et al., 2010*). Positional and isotropic atomic displacement parameter (ADP) as well as TLS ADP refinement of native Imp9•H2A-H2B with NCS restraints was performed to a resolution of 2.70 Å using the *Phenix* program suite with a random 2.1% of all data set aside for an $R_{free}$ calculation. The final model for Imp9•H2A-H2B ($R_{work}$ = 20.9%, $R_{free}$ = 24.0%) contained 2275 residues and 356 waters. The relatively high $R_{work}$ and $R_{free}$ values are likely due to the presence of translational pseudosymmetry. A Ramachandran plot generated with the program *MolProbity* (*Chen et al., 2010*) indicated that 97.1% of all protein residues are in the most favored regions and 0.1% in disallowed regions. Illustrations were prepared with PyMOL (*Schrodinger LLC, 2015*). Data collection and structure refinement statistics are summarized in *Figure 1—source data 1*.

## Quantification of binding affinities by isothermal titration calorimetry (ITC)

Imp9 and mutant Imp9 proteins were expressed and purified as described above. The wild type full-length H2A, H2B and mutant H2BΔ(1-35) proteins were purified as described above. Mutant H2AΔTail and H2BΔTail proteins were obtained from Histone Source (Colorado, USA). H2A-H2B, H2AΔTail -H2B, H2A-H2BΔ(1-35) and H2AΔTail-H2BΔTail heterodimers were reconstituted and purified as described above. Imp9 or mutant Imp9 proteins and H2A-H2B or H2A-H2B mutant dimers were dialyzed in ITC buffer containing 20 mM Tris-HCl (pH 7.5), 150 mM NaCl, 5 mM TCEP and 5% glycerol. ITC experiments were carried out using ITC-200 calorimeter (Microcal, LLC, Northampton, MA, USA) at 20°C with 0.035 mM of Imp9 or mutant Imp9 protein in the sample cell and 0.35 mM H2A-H2B or mutant H2A-H2B protein in the syringe. All samples were thoroughly degassed and then centrifuged at 16000 g for 10 min to remove precipitates. 21 injections each of 1.9 µl except

for the first (0.5 µl) were sequentially made in each experiment. The injections were mixed at 300 rpm and consecutive injections were separated by 300 s to allow the peak to return to baseline. All experiments were carried out in triplicates. Data were integrated and baseline corrected using NITPIC (*Keller et al., 2012*). The baseline corrected and integrated data were globally analyzed in SEDPHAT (*Houtman et al., 2007*; *Zhao et al., 2015*) using a model considering a single class of binding sites. SVD-reconstructed thermogram provided by NITPIC, the fit-isotherms and the residuals from SEDPHAT were all plotted using GUSSI (*Brautigam, 2015*). Individual experiments in the triplicate sets are differently color-coded in *Figure 1—figure supplement 1A*. For error reporting, we used F-statistics and error-surface projection method to calculate the 68.3% confidence intervals of the fitted data (Bevington). The $K_D$ (nM), $\Delta H$ (kCal/mol), $\Delta S$ (Cal/mol.K), $\Delta G$ (kCal/mol) and the Imp9 local concentration correction factors for each set of triplicate experiments are reported in the *Table 1*.

## Pull-down binding assays

Pull-down binding assays were performed by immobilizing purified MBP-Imp9 or MBP-RanGTP (*S. cerevisiae* Gsp1(1–179/Q71L) on amylose resin (New England BioLabs, Ipswich, MA). 40 µl of 100 µM MBP-Imp9 or MBP-RanGTP was immobilized on 200 µl of amylose resin (50% slurry) in binding assay (BA) buffer containing 20 mM HEPES pH 7.3, 110 mM potassium acetate, 2 mM magnesium acetate, 2 mM DTT and 15% glycerol. 100 µl of ~20 µM of immobilized MBP-Imp9 resin was incubated with 100 µl of 400 µM of purified H2A-H2B in a total reaction volume of 400 µl for 30 min at 4°C, followed by five washes each with 400 µl BA buffer. 100 µl of ~20 µM of MBP–RanGTP resin were incubated with 100 µl of 50 µM of purified Imp9•H2A-H2B in a total volume of 400 µl for 30 min at 4°C, followed by five washes each with 400 µl BA buffer.

For RanGTP dissociations assays, a gradient of 10 µl, 20 µl, 40 µl or 60 µl of approximately 500 µM purified RanGTP was added to 50 µl of ~20 µM of immobilized MBP-Imp9 that were pre-bound with H2A-H2B, in a total reaction volume of 400 µl. These binding reactions contain 12.5 µM, 25 µM, 50 µM or 75 µM of RanGTP added to 2.5 µM MBP-Imp9•H2A-H2B. Binding was followed by five washes each with 400 µl of the BA buffer. From each of the reactions, 30 µl of beads after final wash was suspended in 30 µl of BA buffer. 10 µl of the bead slurry sample was analyzed on 12% SDS-PAGE gels and stained with Coomassie Blue for visualization. A control experiment involving immobilized GST-Kapβ2, MBP-PY-NLS (PY-NLS of hnRNP A1), and a gradient of 12.5 µM, 25 µM, 50 µM and 75 µM of RanGTP (prepared as described above for the MBP-Imp9•H2A-H2B experiments) was carried out similarly to show that RanGTP dissociates PY-NLS bound to Kapβ2. 2% of the input RanGTP added in each of the binding reactions and approximately 2% of flow-through is also shown in the Coomassie-stained SDS-PAGE gels.

Pull-down binding assay to probe Ran binding to Imp9 versus Imp9Δ1–144 were performed by immobilizing GST-Imp9 or GST-Imp9Δ1–144 on Glutathione Sepharose 4B resin (GE Healthcare Life Sciences). 12.5 ml of lysate from 500 ml cell culture ($OD_{600}$ = 1) pellet of E. coli expressing GST-Imp9 or GST-Imp9Δ1–144 (containing ~8 mg/ml of GST-Imp9 protein) were incubated on 1 ml of 50% Glutathione Sepharose 4B slurry in BA buffer. The GST-Imp9 or GST-Imp9Δ1–144 bound resin was washed five times, each with 6 ml BA buffer, before the binding assay. 200 µl of 50% slurry GST-Imp9 or GST-Imp9Δ1–144 resin (~12 µM proteins) was incubated with 10 µl of ~500 µM RanGTP in a total reaction volume of 400 µl for 30 min at 4°C, followed by five washes (each with 400 µl BA buffer). After washing, 30 µl of 50% beads slurry was suspended in 30 µl BA buffer. 10 µl of the resulting bead slurry sample was analyzed by Coomassie-stained SDS-PAGE. A control experiment using empty GSH sepharose beads and RanGTP was performed as described above.

## Size Exclusion Chromatography

The interaction between RanGTP and Imp9•H2A-H2B complex was probed by size exclusion chromatography (SEC). Imp9, RanGTP, H2A-H2B were purified as described above. First, a series of SEC experiments titrating RanGTP was performed. SEC of Imp9 alone (20 µM), RanGTP alone (60 µM), H2A-H2B alone (20 µM), Imp9 +H2A-H2B 1:1 molar ratio (20 µM) with no RanGTP, 0.5, 1, 2 and 3 molar equivalents of RanGTP were performed in buffer containing 20 mM HEPES pH 7.4, 200 mM sodium chloride, 2 mM magnesium acetate, 2 mM TCEP and 8% (v/v) glycerol. The experiments were performed using a Superdex 200 Increase 10/300 GL column. A second series of SEC

experiments using 1:1 Imp9 +H2A-H2B (70 μM) and 1:1:1 Imp9, H2A-H2B, and Ran (70 μM) in the same column and the same buffer were performed with higher concentrations of proteins for visualization of proteins in the SEC fractions by Coomassie-stained SDS-PAGE. A third SEC series involves the mutant MBP-Imp9Δ1–144 that does not bind RanGTP and using a different Superdex 200 Increase10/300 GL column with buffer containing 20 mM HEPES pH 7.4, 200 mM sodium chloride, 2 mM magnesium acetate, 2 mM DTT and 10% glycerol.

## Analytical ultracentrifugation

The sedimentation coefficients of individual proteins and protein complexes in the mixture were estimated by monitoring their sedimentation properties in a sedimentation velocity experiment carried out in Beckman-Coulter Optima XL-1 Analytical Ultracentrifuge (AUC). The individual proteins and mixtures of proteins were analyzed in AUC buffer containing 20 mM HEPES pH 7.3, 200 mM sodium chloride, 2 mM magnesium chloride, 2 mM TCEP and 8% glycerol (details below). Protein samples (450 μl) and AUC buffer (450 μl) were loaded into a double sector centerpiece and centrifuged in an eight-hole An-50Ti rotor to 50000 rpm at 20°C. The double sectors were monitored for absorbance at 280 nm ($A_{280}$). A total of 140 scans were collected and the first 80 scans were analyzed. Buffer density, viscosity of the buffer and partial specific volume of the protein was estimated using SEDN-TERP (http://www.rasmb.bbri.org/software/windows/sednterp-philo/). Sedimentation coefficient distributions $c(s)$ (normalized for absorption differences) were calculated by least squares boundary modeling of sedimentation velocity data using SEDFIT program (*Schuck, 2000*). Sedimentation coefficients $sw$ (weighted-average obtained from the integration of $c(s)$ distribution) and frictional ratios $f/f_0$ were obtained by refining the fit data in SEDFIT (*Schuck, 2000*). For error reporting, F-statistics and Monte-Carlo for integrated weight-average $s$ values were used (Bevington). Data were plotted using GUSSI (*Brautigam, 2015*).

Individual proteins, Imp9, RanGTP and H2A-H2B, were purified as described above and dialyzed into the AUC buffer before mixing the samples to the final volume of 450 μL for the AUC experiments. Samples for the AUC experiments contain: 1) 450 μL Imp9 alone (3 μM), 2) 450 μL RanGTP alone (10 μM), 3) 450 μL H2A-H2B (10 μM), 4) 3 μM Imp9 + 3 μM RanGTP in a total volume of 450 μL, 5) 3 μM Imp9 +3 μM H2A-H2B in a total volume of 450 μL, 6) 3 μM Imp9 + 3 μM H2A-H2B + 10 μM RanGTP in a total volume of 450 μL. The proteins were mixed overnight before loading into the AUC cell.

## Native gel shift assays

### Electrophoretic Mobility Shift Assays

One protein component was held constant at 10 μM and the other was titrated. Samples were separated by 5% polyacrylamide gel electrophoresis. Gels were run for 100 min at 150 V at 4°C in 0.5x TBE (40 mM Tris-HCl pH 8.4, 45 mM boric acid, 1 mM EDTA). Gels were stained with Coomassie Blue.

### Competition Assays

Nap1, Imp9 or Imp9-Ran (equimolar Imp9 and RanGTP added together without further purification) were titrated (at 0.5, 1.0 and 1.5 molar equivalents of H2A-H2B) against 147 bp Widom 601 DNA mixed with H2A-H2B at 1:7 (1.5 μM:10.5 μM), or 147 bp Widom 601 DNA was titrated against 10.5 μM Nap1, Imp9 or Imp9-Ran (1:1) pre-mixed with an equimolar amount of H2A-H2B. Samples were separated by 5% polyacrylamide gel electrophoresis. Gels were run for 75 min at 150 V at 4°C in 0.5x TBE. Gels were stained with ethidium bromide and then Coomassie Blue.

### Nucleosome Assays

Tetrasomes containing H3-H4 and 147 bp Widom 601 DNA were reconstituted as described in Dyer et al (*Dyer et al., 2004*). To monitor nucleosome assembly, tetrasomes were held constant at 2.5 μM and H2A-H2B or pre-formed complexes of Nap1-H2A-H2B (1:1), Imp9-H2A-H2B (1:1), or Imp9-H2A-H2B-Ran (1:1:1) were titrated. To monitor nucleosome disassembly, Nap1, Imp9, or Imp9-Ran (1:1) complex was titrated against nucleosomes (2.5 μM). Samples were separated by 5% polyacrylamide gel electrophoresis. Gels were run for 75 min at 150 V at 4°C in 0.5x TBE. Gels were stained with ethidium bromide and then Coomassie Blue.

## Small angle x-ray scattering

SAXS experiments examining Imp9, Imp9•H2A-H2B, Imp9•RanGTP, and RanGTP•Imp9•H2A-H2B samples were carried out at Beamline 4–2 of the Stanford Synchrotron Radiation Lightsource (SSRL) in the SLAC National Accelerator Laboratory. At SSRL, the beam energy and current were 11 keV and 500 mA, respectively. A silver behenate sample was used to calibrate the q-range and detector distance. Data collection was controlled with Blu-Ice (*McPhillips et al., 2002*). We used an automatic sample delivery system equipped with a 1.5 mm-diameter thin-wall quartz capillary within which a sample aliquot was oscillated in the X-ray beam to minimize radiation damage(*Martel et al., 2012*). The sample was placed at 1.7 m from a MX225-HE (Rayonix, USA) CCD detector with a binned pixel size of 292 by 292 μm (*Figure 4—source data 1*).

All protein samples for SAXS were expressed and purified as described above. Purified Imp9 was exchanged into SAXS buffer (20 mM HEPES pH 7.3, 110 mM potassium acetate, 2 mM magnesium acetate, 2 mM DTT, and 10% glycerol) by SEC and concentrated to 5 mg/ml (43 μM of Imp9) for SAXS analysis. The Imp9•H2A-H2B was purified as described above and then exchanged into SAXS buffer by SEC and concentrated to 5 mg/ml (35 μM of Imp9•H2A-H2B) for SAXS analysis. To prepare the Imp9•RanGTP complex, previously purified Imp9 was first mixed with 5-fold molar excess of RanGTP for SEC to separate the Imp9•RanGTP complex from excess Ran. This Imp9•RanGTP complex was then buffer exchanged into SAXS buffer in another round of SEC and concentrated to 5 mg/ml (37 μM of Imp9•RanGTP) for SAXS. To prepare the RanGTP•Imp9•H2A-H2B complex, previously purified Imp9•H2AH2B was mixed with 5-fold molar excess of RanGTP in SAXS buffer for SEC to separate RanGTP•Imp9•H2A-H2B from excess RanGTP. Fractions containing RanGTP•Imp9•H2A-H2B were pooled, concentrated and subjected to a second round of SEC in SAXS buffer, after which the complex was concentrated to 5 mg/ml (31 μM of RanGTP•Imp9•H2A-H2B) for SAXS. The 10% glycerol in the SAXS buffer protects the protein samples from radiation damage during X-ray exposure (*Kuwamoto et al., 2004*) and our early studies show that low glycerol concentrations (5–20%) do not affect protein compaction (*Yoshizawa et al., 2018*). All solutions were filtered through 0.1 μm membranes (Millipore) to remove any aggregates. The SAXS profiles were collected at protein concentrations ranging from 0.5 to 5.0 mg/ml. 20 one-second exposures were used for each sample and buffer maintained at 15°C. Each of the resulting diffraction images was scaled using the transmitted beam intensity, azimuthally integrated by SASTool (*SasTool, 2013*) and averaged to obtain fully processed data in the form of intensity versus q [$q = 4\pi \sin(\theta)/\lambda$, $\theta$ = one half of the scattering angle; $\lambda$ = X ray wavelength]. The buffer SAXS profile was subtracted from a protein SAXS profile. Subsequently, the mean of the lower concentration (0.5–1.5 mg/ml) profiles in the smaller scattering angle region ($q < 0.15$ A$^{\circ -1}$) and the mean of the higher concentration (2.0–5.0 mg/ml) profiles in the wider scattering angle region ($q > 0.12$ A$^{\circ -1}$) were merged to obtain the final experimental SAXS profiles that are free of the concentration-dependent aggregation or polydispersity effect (*Kikhney and Svergun, 2015*).

The merged SAXS profiles were initially analyzed using the ATSAS package (*Petoukhov et al., 2012*) to calculate radius of gyration ($R_g$), maximum particle size ($D_{max}$), and pair distribution function ($P(r)$) (*Figure 4—figure supplement 3* and *Figure 4—source datas 1* and *2*). The molecular weight ($MW_{SAXS}$) of each SAXS sample was estimated using SAXS MOW (*Fischer et al., 2010*) with a threshold of $q_{max} = 0.25$–0.3 Å$^{-1}$ (*Figure 4G* and *Figure 4—source datas 1* and *2*).

## Co-immunoprecipitation and immunoblotting

HeLa cells expressing H2B$^{mCherry}$ (*Ke et al., 2011*) (gift from Prof. Hongtao Yu, UT Southwestern). The HeLa Tet-ON cells (Cellosaurus Accession: HeLa Tet-On (CVCL_IY74)) stably expressing H2B-mCherry were originally created (with cell identity confirmation carried out by STR profiling) in Dr. Hongtao Yu's lab at University of Texas Southwestern Medical Center, Dallas, Texas USA. Mycoplasma negative status of the cell line was confirmed using the LookOut Mycoplasma PCR Detection kit, Sigma MP0035-1KT. The cells were grown to 80% confluency, and total-cell lysate was prepared by suspending the cells in TB buffer containing 20 mM HEPES–KOH pH 7.3, 110 mM potassium acetate, 2 mM magnesium acetate, 5 mM sodium acetate, 0.1 mM EGTA, 1 mM DTT and protease inhibitor cocktail (*Kimura et al., 2017*) on ice for 15 min, sonicating three times (5 s pulse, 10 s rest), then centrifuging the lysed cells at 15,000 g for 20 min at 4°C. Nuclear and cytoplasmic fractions were prepared using the NE-PER Nuclear and Cytoplasmic Extraction reagents (Thermo Scientific)

as per manufacturer's instruction. Protein concentration was quantitated using the Bradford protein assay kit (BioRad). The RFP-Trap (high quality Red Fluorescent Protein (RFP) binding protein coupled to a monovalent magnetic matrix, ChromoTek GmbH) was incubated with the cell lysates for 2 hr at 4°C. The matrix was first washed with TB buffer supplemented with 200 mM NaCl, and then once with TB buffer supplemented with 150 mM NaCl. The proteins bound to the beads were dissolved in SDS sample buffer for immunoblot analysis.

Cell lysate and protein samples dissolved in SDS sample buffer were separated by SDS–PAGE, and blotted with the indicated antibodies: Rabbit polyclonal antibody against tagRFP (1:1000 dilution, Cat no. AB233, Evrogen), rabbit polyclonal antibody against Imp9 (1:1000 dilution, Cat no. A305-474A-T, Bethyl Laboratories, Inc), mouse monoclonal against Ran (1:2000 dilution, Cat no. 610340, BD Biosciences), mouse monoclonal against Nuclear Pore Complex Proteins Antibody [MAb414] (1:5000 dilution, Cat no. 902903, BioLegend) and mouse monoclonal against PCNA (1:2000 dilution, Cat no. 307901, BioLegend). Goat anti-Rabbit IgG (H + L), HRP-conjugated (1:6000 dilution, Cat no. 31460, Thermo Fisher Scientific) and Goat anti-Mouse IgG (H + L), HRP-conjugated (1:6000 dilution, Cat no. 31430, Thermo Fisher Scientific) were used as the secondary antibodies, and immunoblots were developed using the SuperSignal West Pico PLUS Chemiluminescent Substrate (Cat no. 34580, Thermo Fisher Scientific) according to the manufacturer's protocols and followed by detection using a Gel Doc EZ System (Bio-Rad Laboratories, Hercules, CA, USA.

## Confocal microscopy imaging

Cells ($5 \times 10^4$ cells per chamber) were seeded into collagen coated culture coverslip (BD Falcon) The next day, cells were rinsed with ice-cold PBS and fixed with 4% paraformaldehyde for 10 min at room temperature followed by permeabilization with 0.1% sodium citrate plus 0.1% Triton X-100. The cells were subjected to immunofluorescence staining using rabbit polyclonal antibody against Imp9 (1:250 dilution, Cat no. A305-474A-T, Bethyl Laboratories, Inc) and mouse monoclonal antibody against Ran (1:250 dilution, Cat no. 610340, BD Biosciences), for 2 hr at room temperature. The cells were then washed with cold PBS three times for 1 min each and incubated with Alexa 480-labeled anti-rabbit secondary antibody (1:800) (Invitrogen) and Alexa 405-labeled anti-mouse secondary antibody (1:800) (Invitrogen) at room temperature for 1 hr. Subsequently cells were washed with cold PBS three times for 1 min each and mounted with ProLong Gold Antifade Mountant (Invitrogen).

Image acquisition was performed with a spinning disk confocal microscope system (Nikon-Andor) with a 100 × oil lens and the MetaMorph softwar. Images were acquired from randomly selected fields as a z-stack with step size of 0.1 μm to give a total of 196 slices. For each selected field of view, three images were taken, an Alexa488 (Imp9) image, and Alexa405 (Ran).

## Acknowledgements

We thank Bing Li for plasmids expressing H2A and H2B, Hongtao Yu for the H2B$^{mCherry}$ stable cell lines, Binita Shakya for Ran protein, James Chen for advice on X-ray structure determination, and the Structural Biology Laboratory and Macromolecular Biophysics Resource at UTSW for their assistance with crystallographic and biophysical data collection. Crystallographic results are derived from work performed at Argonne National Laboratory, Structural Biology Center at the Advanced Photon Source. Argonne is operated by UChicago Argonne, LLC, for the U.S. Department of Energy (DOE), Office of Biological and Environmental Research (BER) under contract DE-AC02-06CH11357. We thank T Matsui and TM Weiss at SSRL, SLAC National Accelerator Laboratory, for assistance with collecting SAXS data. SAXS experiments were performed at the SSRL, SLAC National Accelerator Laboratory operated for DOE by Stanford University. The SSRL SMBP is supported by the DOE BER, by the National Institutes of Health (NIH), NCRR, Biomedical Technology Program (P41RR001209), and by NIGMS, NIH (P41GM103393). This work was funded by NIGMS of NIH under Awards R01GM069909 (YMC), U01GM98256-01 (YMC and AS), R01GM083960 (AS), P41GM109824 (AS), R01GM112108 (AS), the Welch Foundation Grants I-1532 (YMC), the Leukemia and Lymphoma Society Scholar Award (YMC), start-up funds from the University of Texas at Dallas (SD) and the University of Texas Southwestern Endowed Scholars Program (YMC).

## Additional information

### Funding

| Funder | Grant reference number | Author |
|---|---|---|
| National Institute of General Medical Sciences | U01GM98256-01 | Andrej Sali Yuh Min Chook |
| National Institute of General Medical Sciences | R01GM083960 | Andrej Sali |
| National Institute of General Medical Sciences | P41GM109824 | Andrej Sali |
| National Institute of General Medical Sciences | R01GM112108 | Andrej Sali |
| University of Texas at Dallas | Start-up funds | Sheena D'Arcy |
| National Institute of General Medical Sciences | R01GM069909 | Yuh Min Chook |
| Welch Foundation | I-1532 | Yuh Min Chook |
| Leukemia and Lymphoma Society | Scholar Award | Yuh Min Chook |
| University of Texas Southwestern Medical Center | Endowed Scholars Program | Yuh Min Chook |

The funders had no role in study design, data collection and interpretation, or the decision to submit the work for publication.

### Author contributions

Abhilash Padavannil, Formal analysis, Investigation, Writing—original draft, Writing—review and editing; Prithwijit Sarkar, Tolga Cagatay, Formal analysis, Investigation; Seung Joong Kim, Investigation, Methodology; Jenny Jiou, Investigation; Chad A Brautigam, Formal analysis, Methodology, Writing—review and editing; Diana R Tomchick, Formal analysis, Validation, Methodology, Writing—review and editing; Andrej Sali, Supervision, Funding acquisition, Methodology; Sheena D'Arcy, Formal analysis, Supervision, Funding acquisition, Methodology, Writing—original draft, Writing—review and editing; Yuh Min Chook, Conceptualization, Supervision, Funding acquisition, Methodology, Writing—original draft, Project administration, Writing—review and editing

### Author ORCIDs

Diana R Tomchick (iD) http://orcid.org/0000-0002-7529-4643
Andrej Sali (iD) http://orcid.org/0000-0003-0435-6197
Sheena D'Arcy (iD) http://orcid.org/0000-0001-5055-988X
Yuh Min Chook (iD) http://orcid.org/0000-0002-4974-0726

### Decision letter and Author response

Decision letter https://doi.org/10.7554/eLife.43630.029
Author response https://doi.org/10.7554/eLife.43630.030

## Additional files

### Supplementary files

• Transparent reporting form
DOI: https://doi.org/10.7554/eLife.43630.024

### Data availability

Diffraction data have been deposited in PDB under the accession code 6N1Z.

The following dataset was generated:

| Author(s) | Year | Dataset title | Dataset URL | Database and Identifier |
|---|---|---|---|---|
| Tomchick DR, Chook YM, Pada-vannil A | 2018 | Importin cargo complex | http://www.rcsb.org/structure/6N1Z | Protein Data Bank, 6N1Z |

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
