## [Decision Letter]

[Editors’ note: a previous version of this study was rejected after peer review, but the authors submitted for reconsideration. The first decision letter after peer review is shown below.]

Thank you for submitting your work entitled "Importin-9 wraps around the H2A-H2B core to act as nuclear importer and histone chaperone" for consideration by *eLife*. Your article has been reviewed by three reviewers and the evaluation has been overseen by a Senior Editor. The following individuals involved in review of your submission have agreed to reveal their identity: André Hoelz (Reviewer #1).

Our decision has been reached after consultation between the reviewers. Based on these discussions and the individual reviews below, we regret to inform you that your work will not be considered further for publication in *eLife*.

As you will see, the reviewers praise parts of the work and acknowledge the importance of certain achievements. On the other hand, they also identify several important weaknesses, both in the experimental side and in the interpretation of the results. The prevailing view was that the manuscript, at least in its present form, does not meet the standards expected at *eLife*.

*Reviewer #1:*

Padavannil et al., present the structure of a histone H2A:H2B pair in complex with its previously identified nuclear transport factor Importin-9 (Imp9). Subsequently, the authors probe the role of Imp9 in nucleosome biogenesis leading to the conclusion that Imp9 acts as a bona fide histone assembly chaperone. The manuscript is well written and easy to follow. However, the figures and the experiments lack the attention to detail that one expects from the Chook laboratory. While the findings are of potential broad interest to the readership of *eLife*, the biochemical experiments are insufficient to support key conclusions. There are also additional experiments well within the expertise of the Chook laboratory that would increase the impact of this study. The senior author should carefully re-work the figures and revise the text according to the following points:

Figure 1:

The structure is somewhat unexpected and appears carefully done, with the extensive interface between Imp9 and the H2A:H2B histone pair the most striking feature. It is a great example of the plasticity of β-karyopherins and their ability to recognize cargoes by their shape rather than by a linear nuclear localization sequence. Nevertheless, the statistics reported in Figure 1—source data 1 require major revision (see below). The Imp9 schematic shown in panel B does not convey any additional information required to digest the overall structure and should be moved to the supplement. Panels E-G are difficult to digest, particularly when printed. Labels should be restricted to residues shown as sticks to contact forming positions only. Additionally, a broader box view of each interface presented in the supplement using cross-eye stereo, alongside images of representative electron density would be informative. A sequence alignment of Imp9 homologs, highlighting the residues of the 3 identified H2A:H2B binding interfaces should be incorporated in the supplement demonstrating that this extensive 3-dimensional binding interface is evolutionarily conserved.

Figure 2:

Panel 2A: The cartoon representation of H2A:H2B does not provide useful information. Consider replacing this panel with an overview of the assembled nucleosome, highlighting the H2A:H2B regions where Imp9 binding prevents assembly in the context of intact nucleosomes.

Panel 2C: This experiment should be repeated on a continuous native gel to facilitate direct comparison, please consider showing all pull-down and gel shift assays this way. This experiment also lacks an input demonstrating the quality of the protein components used.

Panel 2D: The molecular weight marker is currently not labeled.

Figure 3:

Panel 3A: Unlike other nuclear import complexes, RanGTP is insufficient to release histone H2A:H2B cargos from Imp9. This experiment is at the heart of the manuscripts novelty and requires further experimental support to sufficiently demonstrate Imp9:H2A:H2B:RanGTP tetramer formation. (1) Loading and input gels, as well as bead controls are missing, each individual component should be tested for unspecific MBP resin binding and these controls should be included in the figure. This issue is common across all the pull-downs presented in the entire manuscript and must be addressed in all cases. (2) The current experiment is a single observation, titrating various RanGTP concentrations to super-stoichiometric levels would demonstrate that this complex is resistant to typical cargo release, particularly in the nuclear environment where RanGTP is plentiful.

Panel 3B: This panel presents a gel shift assay that shows that RanGTP sensitizes the Imp9:H2A:H2B:RanGTP import complex to release the histone pair upon recognition by DNA. This experiment should span a single continuous native gel to allow for direct comparison. Given that 3 replicates were performed, quantitation of the total DNA bound to the Imp9:H2A:H2B:RanGTP heterotetramer should be reported. Furthermore, this data contradicts the concluding statement regarding transfer to alternative histone assembly chaperones, as Imp9 appears capable of directly mediating transfer to nucleosomal DNA. Indeed, it is not intuitive why the Imp9:H2A:H2B:RanGTP complex should be dismantled by double-stranded DNA, as this would expose the hydrophobic H2A:H2B surfaces that interact with H3:H4 in the histone octamer. Instead, for me all data points towards a mechanism whereby a RanGTP primed Imp9:H2A:H2B import complex would serve as a direct H2A:H2B donor for nucleosome assembly. This would be easily testable if the Imp9:H2A:H2B:RanGTP import complex would be mixed with various H3:H4:histone chaperone complexes and histone octamer formation assayed either by pull down or by gel filtration analysis. For example, the H3:H4:Rtt106 complex could be used for this experiment and can be made from bacterially expressed proteins (Su et al., 2012). This assay could also be performed in the presence of the Widom 601 DNA to test whether intact nucleosomes can be assembled. A gel filtration assay would also rule out potential artifacts that are caused by aggregated H2A:H2B pairs in these reported assays.

Panel 3C: The sedimentation coefficient X axis of the analytical ultracentrifugation plot extends too far, it would be useful to focus on a smaller range, perhaps 0-7, to allow for easier identification of the different peaks.

Panel 3D: The nucleotide that is bound to Ran should be labeled in the figure. Furthermore, this panel should clearly be labeled with "MODEL" to prevent readers from concluding that the crystal structure of the RanGTP-bound form of the Imp9:H2A:H2B complex has been determined. The model presumes that RanGTP binds Imp9 in a canonical fashion, modeled on the *S. cerevisiae* Kap121:RanGTP structure, however its binding does not dismantle the import complex.

Panel 3E: The entire SAXS analysis does not build any confidence that the interaction between RanGTP and the Imp9:H2A:H2B complex is modelled correctly. This data can either be omitted entirely from the manuscript or moved to the supplement. Instead, RanGTP binding to the Imp9:H2A:H2B complex should be validated by Imp9 surface mutagenesis, taking advantage of previously determined crystal structures of β-karyopherins bound to RanGTP. A zoomed in view, detailing the presumably conserved binding determinants shared between Imp9 and the previously characterized β-karyopherins in complex with RanGTP should be incorporated into this figure alongside the site-directed mutagenesis analysis. If the model is correct, mutations should be readily identifiable that mutants that abolish the RanGTP interaction. Additionally, the key take home message from this figure is that RanGTP cannot trigger H2A:H2B release but instead alters the import complex to permit handover to DNA. However, the data presented do not sufficiently rule out the possibility that large excess of RanGTP can trigger release, which is the condition in the nucleus. Thus, a gel filtration interaction analysis experiment, preferably coupled to multi-angle light scattering to determine stoichiometries, showing stable RanGTP:Imp9:H2A:H2B complex formation by incubating Imp9:H2A:H2B with equimolar or 10-fold excess of RanGTP prior to injection would demonstrate complex formation and rule out disassembly.

Figure 1—figure supplement 1: The ITC analysis is commendably thorough. However, the titrations conducted in panels D, E and F must be repeated. The injection interval is too short and does not allow the instrument to equilibrate to baseline. Additionally, standard deviations for the dissociation constants should be reported. Have the experiments been carried out multiple times? Also, ΔG, ΔH and ΔS values should be reported.

Figure 1—figure supplement 1B: The surface charge analysis should be moved to a new figure, with the interface presented in an open book format. Bold outlines denoting the location of the 3 interfaces between Imp9 and H2A:H2B are required to orient the reader to the distinct surfaces involved. Additionally, please use and cite APBS for the surface charge calculations.

Figure 1—source data 1:

This is not a standard X-ray crystallography data collection and refinement statistics table. From top to bottom: Cell dimensions should be reported with 1 decimal point (not 127.424 but 127.4), the B-factors should be reported as full numbers (not 52.81 but 53). The resolution is inconsistently reported at 2.7 Å here and 2.65 Å in the methods. Furthermore, a *R_merge_*of 0 and I/σ(I) of 1 in the highest resolution shell, I am convinced that the high-resolution limit has not been determined correctly. The I/σ(I) reported in the PDB validation report is 1.9 contradicting what is reported in the table. While direct detectors allow the usage of all data to background levels, proper resolution criteria should be reported. Thus, please report the CC_1/2_ values and use this parameter to properly determine the high-resolution limit. *R_merge_* should be reported in percent (not 0.085 but 8.5% ). *R_work_* and *R_free_*should only be reported for the entire dataset, omit highest resolution shell. As the structure does not contain ligands or ions, the corresponding atomic and B-factor values can be omitted. 356 water molecules modeled into a structure at 2.7 Å resolution appears unusually high, especially with weak or no data in the highest resolution shell. Were waters modeled into noise? Limit the placement of water molecules to a sensible B-factor. The Molprobity and Ramachandran statistics are missing, these are essential quality control values that must be reported in the table. For both datasets the number of total and unique reflections are missing. Finally, the PDB code is not reported, the authors should formally submit the structure to the PDB and include the accession number in the revised manuscript. Within the methods portion for the crystallization and structure determination both the number of molecules in the ASU and how NCS has been treated during refinement should be detailed.

*Reviewer #2:*

In this study, the authors solved the crystal structure of Importin-9 in complex with the histone H2A-H2B dimer. Importin-9 functions as a nuclear import receptor for various (typically highly basic) substrates. The importin-cargo complex docks to nuclear pore complex (NPC) and is subsequently translocated to nucleoplasmic side, where RanGTP binds to the Importin with high affinity and favors release of cargo into nucleus. The structure of importin-9 complex will be appreciated in the field of nuclear transport research as it is quite different from other known import receptor crystal structures. Apart from solving the structure, the authors examine, from a biophysical perspective, the role of Importin-9 as a histone (H2A-H2B) chaperone.

In general, the structure is a magnificent achievement and the paper is nice. However, the manuscript has several issues that need to be mended:

1) The authors make a strong point in their Discussion section that the histone tails would not be recognized by importin-9, based on the fact that they don't see an electron density for the interaction. Nevertheless, the binding strength drops by substantial factors when either the H2A or the H2B tail is deleted (4.5-fold and 13-fold, respectively). The combined effect of both deletions will cause a 60-fold drop in binding strength. This is certainly not a minor effect. Given that the authors measured the affinities at a higher than physiological ionic strength, the drop will be even greater under import conditions. Dismissing the tails' contribution is therefore a misinterpretation.

2) The authors should provide explanations for the missing electron densities. Obvious ones are: (A) that the precipitant used for crystallization is of very high ionic strength (3M potassium formate) and therefore likely to artificially break otherwise important salt bridges and (B) that the tails might engage in a fuzzy interaction with the importin. This is an established concept describing interactions of disordered regions with a globular protein. In the specific case, there are probably several possibilities for how the basic tails salt-bridge to negatively charged regions of the importin.

3) In the light of the just made points, I regard the comments on the Muhlhausser, (2001) paper (Discussion section) as inappropriate.

4) Likewise, the authors give the impression as if they had contradicted the concept of importins functioning as chaperones for highly basic cargoes (Discussion section). The concept (Jäkel et al., 2002) implied that importins shield histones and ribosomal proteins against aggregation with polyanions and gave an intuitive explanation as to why import signals are not always as short as an SV40 NLS. It is unclear why the authors phrase their data as a contradiction. It would be more appropriate to give proper credit; the importin-chaperone concept should already be introduced in the Introduction.

5) The referencing in the introduction is inappropriately biased. For example, there is no mentioning of the first discovered import pathway for a histone (Jakel, 1999). Likewise, the by now established fact that the RanGTPase system determines the direction of transport is referenced by two reviews of the Chook lab – despite the fact that the RanGTPase gradient concept had been worked out by a set of papers by the Görlich lab (see e.g. Izaurralde el al., 1997). Please amend this and similar problems.

6) I have little faith in the model of the RanGTP/ importin-9 complex. The Chook lab published a similar prediction before, namely a model of how RanGTP binds to the CRM1/ snurportin 1 complex (Dong et al., 2009). The experimental structure of the RanGTP/CRM1/ snurportin complex (Monecke et al., 2009), however, revealed that the model was not correct. I cannot see why a prediction of the RanGTP/ importin-9 prediction would be any better. There is less than 20% sequence identity to the most similar importin/ exportin with a solved RanGTP structure. Furthermore, the mode of RanGTP-binding is only partially conserved amongst the members of the importin β superfamily. Some conservation is evident in the contact regions of HEAT repeats 1-3. The additional contacts, however, are not conserved; they differ widely between the various solved RanGTP complexes, and I cannot see how one could possibly predict new ones in any reliable manner. A validation of the modeled interaction by mutagenesis will also not help because such validation will not assess the most problematic aspects of the model, namely contacts that have not been predicted. The RanGTP/ importin-9 structure model should be omitted because there is no reason to believe that it is reliable.

7) The signal to noise ratio in the ITC experiments is rather poor – resulting in rather broad confidence intervals of the fits. Some plots e.g. Figure 1—figure supplement1D, E, G, H failed to reach saturation. Please supply a substantially improved dataset. Since the impact of ionic interactions and the magnitude of the effects are important for the argument, a dataset at physiological salt should be included. For measuring weak interactions, it will help to raise the concentration of the titrant to 30-50-fold higher than the analyte.

8) Please identify the used H2A and H2B variants by including the relevant UniProt numbers.

9) Figure 1A (upper panel). Please identify the protein bands.

10) Figure 1B, please improve clarity of the cartoon.

11) Figure 1E-G are too crowded to get the information. Please re-label to improve clarity, and please label only what is really needed for argument.

12) Please clarify which form of Ran (wt vs. Q69L, yeast of human) is used for which experiment.

13) Fitting of the models into SAXS data is not very conclusive. How well do alternate coordinate models fit the SAXS data?

14) The term "NLS" is ambiguous and by many (if not most) cell biologists considered to be an importin α-dependent nuclear import signal. I would therefore refrain from calling the importin 9-interacting interface an NLS. The more generic term "import signal" should cause less confusion.

*Reviewer #3:*

The manuscript by Padavannil and colleagues (team of Yuh Min Chook and collaborators) reports the crystal structure of human Importin 9 in complex with *X. laevis* histones H2A-H2B to 2.7A resolution. The 20 HEAT repeats of the importin wrap around one H2A-H2B heterodimer, burying a large surface through three surface regions. Some interaction occurs with a C-terminal segment of the N-terminal H2B tail, but otherwise most contacts are mediated between the H2A-H2B core fold and the importin. Next, the authors probe the relevance of specific contacts using isothermal titration calorimetry and deletion mutants. Some limited analysis is used to compare how Nap1 and Imp9 bind H2A-H2B and displace DNA from the H2A-H2B•DNA complex, confirming the ability of Imp9 of shielding histones from inappropriate DNA contacts. Preliminary pulldown experiments between an immobilized MBP-Imp9 fusion constructs and multiple H2A isoforms indicate a broad range of H2A-H2B isoforms interacting with Imp9. Finally, a strong range of biochemical and biophysical assays, including SAXS experiments, indicate that RanGTP binds the Imp9•H2A-H2B complex, rather than leading to the release of the H2A-H2B cargo, similar to what had been observed in pioneering work with the yeast importin and H2A-H2B by Mosammaparast et al., 2002 (which has been duly cited by the authors).

Overall, the reported in the manuscript are of interest to the histone chaperone field. Moreover, the confirmation of the atypical complex between an importin, its cargo and RanGTP simultaneously is of wider molecular interest. The authors also go to a significant level of analysis to confirm that the Imp9-H2A/H2B-RanGTP complex can be formed in vitro. The previous report of a related interaction for the yeast orthologues somewhat dampens the novelty of this specific finding, yet the structure of the Imp9-H2A-H2B complex is important.

Imp9 does not bind the NLS like sequences on the histone tail but covers its surface. This deserves a bit more discussion and a more rigorous analysis, including in ITC assays, as will be suggested below.

Additional experiments to test the existence of the Imp9-H2A-H2B complex in both cytoplasm and nucleus would strengthen the reported interaction at the level of IP in HeLa cell extracts.

ITC data:

- The results appear to have been derived from single measurements, see Figure 1—figure supplement 1A. As can be seen from this example, there are anomalies in data points in the important region of the titration where the heat of injection changes most. This indicates that the data point for full-length H2A-H2B cannot be trusted. Figure 1—figure supplement 1D, E, and F suffer from similar problems. I would urge the authors to conduct these assays in triplicate. In the absence of these data, the comparison between the distinct H2A-H2B constructs appears premature. I encourage the authors also to provide a stoichiometry value, as well as the ΔH and ΔS parameters from the fitting of the results.

- The ΔH18-19 loop deletion construct shows a quite distinct ITC profile, the fitting here was carried out using an endothermic reaction, rather than fitting exotherms. This is not discussed in the manuscript, but ought to briefly be explained.

- Point mutants might also be quite instructive compared to the rather less surgical deletion of entire histone tails and loops, even if their impact to the overall binding energy may be limited.

(Note: considering the aforementioned limitations on the accuracy of the ITC experiments, providing four significant figures appears unjustified)

Pulldown experiment between importin 9 and H2A-H2B constructs:

- Figure 2D lacks an appropriate negative control, such as an unrelated protein, MBP alone and/or an important mutant/distinct isoform that does not bind H2A-H2B.

-The authors claim that the Imp9-H2A-H2B complex is present in the cytoplasm and nucleus and possibly also on chromatin. However, they do not directly show this. Co-IPs in these different cellular fractions would confirm this and would be a nice addition to the paper.

- Additional assays/evidence, such as imaging, showing that RanGTP can bind with Imp9-H2A/H2B in vivo would strengthen the manuscript.

Gel shift assay in Figure 3B:

- As it stands, the current assay is difficult to interpret, as samples are not shown comparable on the same gel. In addition, the conditions that contain Imp9 are different, one is a preformed complex and the other is not. This also makes it difficult to compare activity and make firm conclusions based on this preliminary experiment. This can be readily experimentally addressed.

Figure 4—figure supplement 1:

- There seems to be less H2A/H2B when RanGTP is added. This experiment should be shown on one gel to compare conditions properly. In addition, this referee would suggest adding more RanGTP to test whether H2A-H2B levels decrease further.

[Editors’ note: what now follows is the decision letter after the authors submitted for further consideration.]

Thank you for submitting your article "Importin-9 wraps around the H2A-H2B core to act as nuclear importer and histone chaperone" for consideration by *eLife*. Your article has been reviewed Detlef Weigel as the Senior Editor, a Reviewing Editor, and two reviewers. The reviewers have opted to remain anonymous.

The Reviewing Editor has drafted this decision to help you prepare a revised submission.

Summary:

In this submission of a manuscript that was previously reviewed at *eLife* and has now been resubmitted in a revised form, Padavannil and co-authors have adequately addressed the following concerns:

1) The crystallographic table and crystal structure are now of publishable quality.

2) The manuscript text and figures are now largely of publishable quality.

3) The requested nucleosome assembly/disassembly experiments have been carried out.

4) The revised ITC data are of substantially improved and are now of publishable quality.

5) The RanGTP binding site has been mapped to the N-terminal three HEAT repeats of Imp9.

As already pointed out during the first round of review, the manuscript reports a very interesting set of observation and this new submission is further improved. There are, however, some important issues, raised by reviewer 1 and discussed with the others, that require further consideration.

Essential revisions:

We judge that there is not yet sufficient evidence that the RanGTP:Imp9:H2A:H2B complex is stable in solution, raising the question whether the proposed mechanism is correct that RanGTP binding to the Imp9:H2A:H2B complex is required for priming of the import complex that it can serve as a H2A:H2B donor during nucleosome assembly. Therefore, my criticisms of the revised manuscript refer primarily to the insufficient quality of newly included size-exclusion chromatography (SEC) interaction analysis that was introduced to support the formation of a stable RanGTP:Imp9:H2A:H2B tetramer.

Figure 4—figure supplement 1D. We had requested that the authors carry out additional SEC interaction experiments, ideally combined with a multi-angle light scattering analysis, that would clearly establish that RanGTP binding indeed results in the formation of a heterotetrameric RanGTP:Imp9:H2A:H2B complex, rather than triggering the release of the bound H2A:H2B import cargo and the formation of a heterodimeric Imp9:RanGTP complex, as it has been observed in countless other canonical import complex cases. This analysis is at the heart of the novelty of the presented manuscript. While the authors have included such a SEC analysis, the results of the experiment are uninterpretable. Either the quality of the experiment is insufficient, or the experiment is contrary to the author's conclusion.

The two SDS-PAGE gels of the preincubation run fractions of a mixture of Imp9:H2A:H2B with a 4-fold molar excess of RanGTP do not show stochiometric incorporation of RanGTP into the Imp9:H2A:H2B complex. As judged by the gel filtration profile of RanGTP in isolation, RanGTP elutes as two peaks, consistent with its known weak dimerization behavior: a major peak at 18.2 ml and a second much smaller peak at ~14 ml. The second smaller peak is close to the position where the Imp9:H2A:H2B heterotrimer elutes (12.9 ml). While there is a faint RanGTP band in the SDS-PAGE gel of the preincubation run (lanes 13-16), the band is clearly much weaker than the H2A:H2B bands, indicating at best a sub-stochiometric incorporation of RanGTP into the Imp9:H2A:H2B complex. However, it seems that the faint RanGTP band may not be the result of the incorporation of RanGTP into the Imp9:H2A:H2B complex, but is rather the secondary peak of RanGTP co-eluting independently. An SDS-PAGE gel of the RanGTP control injection is essential to distinguish these two possibilities. Moreover, there is a shift of ~0.4 ml in the elution profiles of RanGTP in isolation compared to the preincubation run profile (this is best seen in the blue and brown major peaks of RanGTP at an elution volume of ~18 ml). Such differences are common when experiments are carried out on different gel filtration columns, with different tubing lengths, or on different FPLC instruments. Additionally, the amounts of the injected proteins are unequal in the various injections and the baselines indicate that the column was not properly equilibrated between different injections. To definitively conclude whether RanGTP stoichiometrically incorporates into the Imp9:H2A:H2B complex, a proper SEC analysis needs to be carried out in which the various components are injected subsequently on the same gel filtration setup. High quality SDS-PAGE gels for all runs are essential for the interpretation of the experiment and should be included in the revised version of the manuscript.

Unfortunately, the poor quality of the newly included SEC analysis casts further doubts on the validity of the SAXS analysis of the RanGTP:Imp9:H2A:H2B complex. At a minimum the current SEC analysis seems to indicate that the RanGTP:Imp9:H2A:H2B heterotetramer is non-stochiometric at the tested Imp9:H2A:H2B concentration of 100 μM. The maximum reported concentration of the RanGTP:Imp9:H2A:H2B complex in the SAXS analysis was 30 μM. Because no stochiometric RanGTP:Imp9:H2A:H2B tetramer was observed in SEC analysis at an injected concentration of 100 μM, which is comparable to the SAXS analysis, if one considers that injected samples are typically ~4-fold diluted on a gel filtration column, it seems unlikely that the SAXS analysis was carried out with a monodisperse sample of the RanGTP:Imp9:H2A:H2B heterotetramer. Unfortunately, the methods provide no information how the authors reconstituted the RanGTP:Imp9:H2A:H2B tetramer for their SAXS analysis. Generally, monodispersity and the formation of a stable species is essential for a meaningful SAXS analysis. Therefore, SAXS experiments are typically carried out either directly following elution from a gel filtration column or better yet by directly coupling the gel filtration column to the SAXS cell.

The observed difference of the major sedimentation coefficient peaks in the AUC analysis of the Imp9:H2A:H2B complex in the absence and presence of a 3-fold molar excess of RanGTP seems to indicate heterotetramer formation. However, AUC analyses are typically performed at much lower concentrations than gel filtration interaction analyses (the authors also provide here no detail in the methods about their employed concentrations). While the pull-down and AUC data seem to support the interpretation that RanGTP is indeed incorporated into the Imp9:H2A:H2B complex without releasing the H2A:H2B cargo, I am puzzled why the gel filtration data does not support this conclusion.

The recommendation is that the authors carry out a proper gel filtration interaction analysis as outlined above. As an important control, the SEC interaction analysis should also include the truncated Imp9:H2A:H2B complex that the authors identified to be deficient in RanGTP binding. Additionally, the entire SAXS analysis should be removed from the manuscript and the structural characterization of the RanGTP:Imp9:H2A:H2B complex and the likely associated conformational change upon RanGTP binding should be published separately in a future study. Alternatively, an EM analysis could be included but I do not think this is necessary.

---

## [Author Response]

[Editors’ note: the author responses to the first round of peer review follow.]

Reviewer #1:[…] Figure 1:The structure is somewhat unexpected and appears carefully done, with the extensive interface between Imp9 and the H2A:H2B histone pair the most striking feature. It is a great example of the plasticity of β-karyopherins and their ability to recognize cargoes by their shape rather than by a linear nuclear localization sequence. Nevertheless, the statistics reported in Figure 1—source data 1 require major revision (see below).

We performed additional rounds of structure refinement and made major revisions to Figure 1—source data 1.

The Imp9 schematic shown in panel B does not convey any additional information required to digest the overall structure and should be moved to the supplement.

We moved the old Figure 1B to the supplement and enlarged it for clarity. It is now Figure 1—figure supplement 1A.

Panels E-G are difficult to digest, particularly when printed. Labels should be restricted to residues shown as sticks to contact forming positions only. Additionally, a broader box view of each interface presented in the supplement using cross-eye stereo, alongside images of representative electron density would be informative.

To improve clarity of the old Figures 1E-G, we have

1) Enlarged each figure panel.

2) Removed many labels of side chains that are partially hidden, thereby relieving the crowding.

These figure panels are now the new Figure 2B-D.

We also added stereo figures of broader views for each of the three interfaces to the supplement. These new figure panels are now in Figure 2—figure supplement 1A-C.

We also added images showing omit map electron density of loops (loop residues were omitted to calculate the omit maps) in each of the three interfaces. These new figures are now in Figure 2—figure supplement 2A-D.

A sequence alignment of Imp9 homologs, highlighting the residues of the 3 identified H2A:H2B binding interfaces should be incorporated in the supplement demonstrating that this extensive 3-dimensional binding interface is evolutionarily conserved.

We added sequence alignment of Imp9 homologs (human, *X. laevis, D. melanogaster* and *S. cerevisiae*) in the Imp9 regions that make up Interfaces 1-3 to the new Figure 2—figure supplement 3A-C. Imp9 residues that contact histones are clearly marked. Residues in the three interfaces are mostly conserved, with Interface 3 being the most conserved. The level of conservation, especially in Interface 1, reflect the use the main chain of many Imp9 residues for interactions with histone residues. We mention this in the figure legend. We also show contacts to Imp9 main chain in Interface 1 in Figure 2—figure supplement 1D. We mention the prevalence of main-chain interactions in manuscript (Results section) where we also refer to Figure 2—figure supplement 1D and Figure 2—figure supplement 3A.

Figure 2:Panel 2A: The cartoon representation of H2A:H2B does not provide useful information. Consider replacing this panel with an overview of the assembled nucleosome, highlighting the H2A:H2B regions where Imp9 binding prevents assembly in the context of intact nucleosomes.

We replaced the old Figure 2A with two 90° views of the nucleosome (new Figure 3A). The nucleosome on the right has one of its H2A-H2B in a similar orientation as the right view of the Imp9-bound H2A-H2B in the new Figure 3B. This way, readers can easily examine Imp9 contact regions (colored dark blue in Figure 3B) relative to the position of H2A-H2B in the nucleosome (Figure 3A). This pair of cartoon representations nicely complement surface representations of H2A-H2B in Figure 3C that compare Imp9, DNA and H3-H4 interfaces.

Panel 2C: This experiment should be repeated on a continuous native gel to facilitate direct comparison, please consider showing all pull-down and gel shift assays this way. This experiment also lacks an input demonstrating the quality of the protein components used.

The gel in the old Figure 2C was in fact a continuous native gel stained with ethidium bromide. It is now the new Figure 3D. We added labeling and revised the figure legend to make this clear. We added a new Figure 3E, which is the Coomassie-stained native gel of the same experiment in Figure 3D. The gel in Figure 3D shows the DNA while the gel in Figure 3E shows the protein components of the experiment.

Panel 2D: The molecular weight marker is currently not labeled.

We labeled all molecular weight markers in Figure 1A, Figure 4A-C and in all relevant figure supplements.

Figure 3:Panel 3A: Unlike other nuclear import complexes, RanGTP is insufficient to release histone H2A:H2B cargos from Imp9. This experiment is at the heart of the manuscripts novelty and requires further experimental support to sufficiently demonstrate Imp9:H2A:H2B:RanGTP tetramer formation. (1) Loading and input gels, as well as bead controls are missing, each individual component should be tested for unspecific MBP resin binding and these controls should be included in the figure. This issue is common across all the pull-downs presented in the entire manuscript and must be addressed in all cases.

We added a gel (new Figure 4—figure supplement 1C) that shows controls of Imp9, histones and RanGTP added to MBP immobilized on amylose resin. None of these proteins bind non-specifically to MBP or to the amylose resin.

For the gels in the new Figures 4A and 4B, we show the inputs and also the flowthrough (FT) materials of the pull-down experiments in the new Figure 4—figure supplement 1A, B. Both the inputs and FTs show the large molar equivalents (5-30) of the RanGTP used.

(2) The current experiment is a single observation, titrating various RanGTP concentrations to super-stoichiometric levels would demonstrate that this complex is resistant to typical cargo release, particularly in the nuclear environment where RanGTP is plentiful.

We repeated the experiment by titrating increasing amounts of RanGTP, starting with 5 molar equivalent of RanGTP (to the MBP-Imp9•H2A-H2B) in lane 6 of the new Figure 4A followed by 10 molar equivalents in lane 7, 20 molar equivalents in lane 8 and 30 molar equivalents of RanGTP in lane 9. Each of the four lanes shows similar amounts of H2A-H2B and RanGTP bound to immobilized Imp9. Little H2A-H2B is displaced from the beads in the FT (lanes 24-27 of the gel in the new Figure 4—figure supplement 1A).

The same 5, 10, 20 and 30 molar equivalents of RanGTP were also used in the control experiment of MBP-PY-NLS binding to GST-Kapβ2 (Figure 4—figure supplement 1B). In this case, even the lowest concentration with 5 molar equivalent of RanGTP easily released all MBP-PY-NLS from immobilized Kapβ2.

Panel 3B: This panel presents a gel shift assay that shows that RanGTP sensitizes the Imp9:H2A:H2B:RanGTP import complex to release the histone pair upon recognition by DNA. This experiment should span a single continuous native gel to allow for direct comparison.

We included continuous native gels showing that DNA is better able to compete for histones from Ran•Imp9•H2A-H2B than from Imp9•H2A-H2B (new Figure 5A and B). We show the Coomassie-stained native gel to report on the protein complexes in Figure 5A. The ethidium bromide gel of the same experiment, reporting on DNA•H2A-H2B and free DNA, is shown in Figure 5B.

Given that 3 replicates were performed, quantitation of the total DNA bound to the Imp9:H2A:H2B:RanGTP heterotetramer should be reported.

We are not sure what DNA the reviewer is referring to. The RanGTP•Imp9•H2A-H2B heterotetramer does NOT bind DNA. Lanes 8-10 of Figure 5A and 5B show that bound RanGTP facilitates release of H2A-H2B from Imp9 to DNA. This is in contrast to lanes 57 where it is clear that when H2A-H2B is bound to Imp9 without RanGTP, the histones were not released to DNA.

Furthermore, this data contradicts the concluding statement regarding transfer to alternative histone assembly chaperones, as Imp9 appears capable of directly mediating transfer to nucleosomal DNA. Indeed, it is not intuitive why the Imp9:H2A:H2B:RanGTP complex should be dismantled by double-stranded DNA, as this would expose the hydrophobic H2A:H2B surfaces that interact with H3:H4 in the histone octamer. Instead, for me all data points towards a mechanism whereby a RanGTP primed Imp9:H2A:H2B import complex would serve as a direct H2A:H2B donor for nucleosome assembly. This would be easily testable if the Imp9:H2A:H2B:RanGTP import complex would be mixed with various H3:H4:histone chaperone complexes and histone octamer formation assayed either by pull down or by gel filtration analysis. For example, the H3:H4:Rtt106 complex could be used for this experiment and can be made from bacterially expressed proteins (Su et al., 2012). This assay could also be performed in the presence of the Widom 601 DNA to test whether intact nucleosomes can be assembled. A gel filtration assay would also rule out potential artifacts that are caused by aggregated H2A:H2B pairs in these reported assays.

We took suggestions from reviewer #1 and reviewer #3 to probe the functions of Imp9•H2A-H2B and RanGTP•Imp9•H2A-H2B using established nucleosome assembly assays. We show in the new Figure 5E that RanGTP•Imp9•H2A-H2B enhances the donation of H2A-H2B to preassembled tetrasome to form nucleosome, but Imp9•H2A-H2B is not able to donate H2A-

H2B to the tetrasome. These results are consistent with our suggestion that the Ran•Imp9•H2AH2B complex is likely the relevant complex to deliver H2A-H2B in the nucleus.

We also performed nucleosome disassembly assays (new Figure 5F). Consistent with results of the assembly assays, Imp9 alone is quite efficient at disassembling nucleosome but the presence of RanGTP prevents it from doing so.

Panel 3C: The sedimentation coefficient X axis of the analytical ultracentrifugation plot extends too far, it would be useful to focus on a smaller range, perhaps 0-7, to allow for easier identification of the different peaks.

We changed the extent of the X-axis of the AUC plot to 0 – 6.5 in the new Figure 4F.

Panel 3D: The nucleotide that is bound to Ran should be labeled in the figure. Furthermore, this panel should clearly be labeled with "MODEL" to prevent readers from concluding that the crystal structure of the RanGTP-bound form of the Imp9:H2A:H2B complex has been determined. The model presumes that RanGTP binds Imp9 in a canonical fashion, modeled on the S. cerevisiae Kap121:RanGTP structure, however its binding does not dismantle the import complex.

We considered reviewer #2’s warning that the Ran•Imp9•H2A-H2B cannot be modeled reliably and took the reviewer’s suggestion to remove the model from the main figure. However, we performed extensive comparative structural analysis of the binding of the switch 1 and 2 regions of RanGTP to HEAT repeats 1-4 of Impβ, Kap95, Kapβ2, Kap121, TrnSR and Importin-13 (see Figure 4—figure supplement 3A-F and Figure 4—figure supplement 4A-F). The locations of Ran binding at these N-terminal HEAT repeats are clearly conserved. So are the Ran sites on HEAT repeats 1-4 of exportins (data not shown). The equivalent Ran-binding site at the Nterminus of Imp9 is predicted by superimposing the 1^st^ four HEAT repeats of Imp9•H2A-H2B with the 1^st^ four HEAT repeats of the Kap121•RanGTP complex (Figure 4—figure supplement 3G). The key observation here is that the predicted location of the Ran site at the N-terminus of Imp9 is separate from the nearby H2A-H2B binding site, suggesting that this potential landing pad for Ran is likely available when Imp9 is bound to H2A-H2B.

Panel 3E: The entire SAXS analysis does not build any confidence that the interaction between RanGTP and the Imp9:H2A:H2B complex is modelled correctly. This data can either be omitted entirely from the manuscript or moved to the supplement.

We removed the SAXS model. But, independent of the SAXS model, the MW weights of the complexes obtained from SAXS experiments show that RanGTP is indeed bound to Imp9•H2AH2B – we kept this data (upper portion of old Figure 3E) as the new Figure 4G.

Instead, RanGTP binding to the Imp9:H2A:H2B complex should be validated by Imp9 surface mutagenesis, taking advantage of previously determined crystal structures of β-karyopherins bound to RanGTP.

We show sequence conservation within the first four HEAT repeats of 6 different importins in a sequence alignment in Figure 4—figure supplement 4G. HEAT repeats 1-4 of the 7 importins are of much more conserved (mean pairwise sequence identity of 20%) than for all the HEAT repeats (mean pairwise sequence identity of 17%). It is also clear in Figure 4—figure supplement 3A-F that the locations of the Ran binding site at the N-terminal HEAT repeats of Impβ, Kap95, Kapβ2, Kap121, Trn-SR2 and Importin-13 are also very conserved.

Examination of details in the Ran binding sites (Figure 4—figure supplement 4A-E) and the sequence alignment of HEAT repeats 1-4 of the importins (Figure 4—figure supplement 4G) show that conservation of contact residues is more structural/positional rather than of the sequence. This is explained in detail in the figure legend:

“G. Sequence alignment of residues in HEAT repeats 1-4 of Imp9, Kap95, Kap121, Importin-β, Importin-13 and Transportin-SR2. Importin positions with identical amino acids are shaded red, and those with conserved amino acids are shown in boxes. […] The majority of Imp9 side chains in the most common/structurally conserved Ran contact sites (marked with black circles) are the same as or have similar chemical characteristics as at least one of the five other importin side chains in that same position, supporting the prediction that RanGTP will likely contact Imp9 at the same location as shown in A-E, on the B-helices of HEAT repeats 1-4.”

From this analysis, one wonders if the 8 most common Ran contact positions that also show sequence conservation are the ones that contribute the most to binding energy. However, this had not been previously tested for any of the previously characterized importin-ran complexes and we feel it is not wise to test these 8 sites on Imp9, without support from an Imp9•RanGTP structure. The only previously reported Importin mutant that decreased Ran binding is an N-terminal truncation mutant of Impβ (Kutay et al., 1997). We made a similar deletion mutant by removing residues 1-144 (HEAT repeats 1-3) of Imp9. Imp9Δ1-144 shows decreased binding to RanGTP (Figure 4—figure supplement 3H).

A zoomed in view, detailing the presumably conserved binding determinants shared between Imp9 and the previously characterized β-karyopherins in complex with RanGTP should be incorporated into this figure alongside the site-directed mutagenesis analysis. If the model is correct, mutations shoud be readily identifiable that mutants that abolish the RanGTP interaction.

We show in Figure 4—figure supplement 3A-F the common location of RanGTP binding at the N-termini (HEATs 1-4) of six different Importins. We compared these sites with the equivalent site on Imp9, which was predicted by structurally aligning HEAT repeats 1-4 of the Kap121•RanGTP complex with Imp9 (Figure 4—figure supplement 3G). In the figure that shows this structural alignment, we do not show the C-terminal half of the Imp9•H2A-H2B structure so as not to imply how that half of Imp9 might interact with RanGTP; we do not know the extent of how the superhelix of Imp9 might rearrange upon binding RanGTP. In Figure 4—figure supplement 3A-F, we observed that the superhelices/solenoids of different Importins adopt variable pitches thereby placing different parts of the C-terminal halves of these Importins to interact with the α-4 helix and basic patch of RanGTP.

We show zoomed-in views of interactions at the Ran-binding sites of several importins in the new Figure 4—figure supplement 4A-E. We also show the side chains in the equivalent site on Imp9 (new Figure 4—figure supplement 4F).

Additionally, the key take home message from this figure is that RanGTP cannot trigger H2A:H2B release but instead alters the import complex to permit handover to DNA. However, the data presented do not sufficiently rule out the possibility that large excess of RanGTP can trigger release, which is the condition in the nucleus. Thus, a gel filtration interaction analysis experiment, preferably coupled to multi-angle light scattering to determine stoichiometries, showing stable RanGTP:Imp9:H2A:H2B complex formation by incubating Imp9:H2A:H2B with equimolar or 10-fold excess of RanGTP prior to injection would demonstrate complex formation and rule out disassembly.

We show gel filtration analysis of a large excess of RanGTP (12 molar equivalent of Ran to Imp9•H2A-H2B) added to Imp9•H2A-H2B in Figure 4—figure supplement 1D. All H2A-H2B elute in the peak at 12.8 ml along with RanGTP and Imp9. No histones are observed at higher elution volume, suggesting no dissociation from Imp9 by RanGTP.

We feel strongly that sedimentation velocity and SAXS analyses are far more rigorous compared to gel filtration for determining if RanGTP indeed forms a complex with Imp9•H2AH2B and for probing the molecular weight of the resulting assembly. The gel filtration data serves as the third mode of support (in addition to AUC and SAXS) to the two pull-down binding assays in Figure 4A and 4C to show formation of the heterotetrameric complex. We also added native gel data showing that a larger complex is formed in solution when RanGTP is added to Imp9•H2A-H2B (new Figure 4D and 4E). This data is yet another (the fourth one) independent support for the formation of a stable RanGTP•Imp9•H2A-H2B heterotetramer.

Figure 1—figure supplement 1: The ITC analysis is commendably thorough. However, the titrations conducted in panels D, E and F must be repeated. The injection interval is too short and does not allow the instrument to equilibrate to baseline.

We fixed the problem with this figure. We took all three reviewers’ concerns about our old ITC data seriously and repeated all ITC experiments. The new ITC experiments were performed more rigorously with much higher quality proteins (gels, gel filtration traces and mass spectrometry data shown in this response document) in buffers with 150 mM NaCl, and the data are consistently cleaner. We used single batches of the different proteins to decrease variations that may arise from different preparations of protein reagents. Every set of interaction was evaluated in triplicate experiments. The new data are summarized in the new Table 1 and the thermograms, traces and fits for the triplicate experiments shown in Figure 1—source data 1.

Additionally, standard deviations for the dissociation constants should be reported. Have the experiments been carried out multiple times? Also, ΔG, ΔH and ΔS values should be reported.

We provide something far better than standard deviations – we provide the 68.3% confidence interval for K_D_ and for ΔH determined by global fit analysis of the triplicates in each experimental set. We added ΔH, ΔS and ΔG, values to the new Table 1.

Figure 1—figure supplement 1B: The surface charge analysis should be moved to a new figure, with the interface presented in an open book format. Bold outlines denoting the location of the 3 interfaces between Imp9 and H2A:H2B are required to orient the reader to the distinct surfaces involved. Additionally, please use and cite APBS for the surface charge calculations.

We added open-book views of each of the surface views of Imp9 and H2A-H2B in the figure. These new views are now shown in the bottom panel of the new Figure 1—figure supplement 1B.

Figure 1—source data 1:This is not a standard X-ray crystallography data collection and refinement statistics table. From top to bottom:

We performed additional rounds of structure refinement. A revised and improved crystallographic statistics table now named Figure 1—source data 1, a PDB validation report and a new methods description for the X-ray crystallographic studies (Materials and methods section) have been provided.

Cell dimensions should be reported with 1 decimal point (not 127.424 but 127.4), the B-factors should be reported as full numbers (not 52.81 but 53).

As the PDB reports unit cell parameters and *B-*factors to two decimal places (please see the revised PDB validation report), we decided to follow the PDB standard format.

The resolution is inconsistently reported at 2.7 Å here and 2.65 Å in the methods.

Due to low percentage completeness for the resolution shell 2.7 – 2.65 Å, we adjusted the high-resolution limit of the data scaling and the final model refinement to 2.70 Å, which caused no observable change in the electron density map or refinement statistics. Please see the revised Figure 1—source data 1 and the Materials and methods section.

Furthermore, a R_merge_ of 0 and I/σ(I) of 1 in the highest resolution shell, I am convinced that the high-resolution limit has not been determined correctly. The I/σ(I) reported in the PDB validation report is 1.9 contradicting what is reported in the table. While direct detectors allow the usage of all data to background levels, proper resolution criteria should be reported. Thus, please report the CC_1/2_ values and use this parameter to properly determine the high-resolution limit.

The <I/σ(I)> as reported in the PDB validation report (1.8) for the last resolution shell is calculated from intensities estimated from amplitudes (i.e., the structure factors in the mmcif file from the final round of refinement). The <I/σ(I)> as reported in our table (1.0) comes directly from the scaling logfile as output by *HKL-3000*, and is calculated from the actual intensities. We are not surprised that there is a slight difference between the two values, given the different sources of the intensity values for the calculation.

This X-ray data was collected on a standard CCD detector, an ADSC Quantum315R detector, and not a direct detector. Due to the very high multiplicity of the native data set (10.1 overall, 8.2 for the highest resolution shell), the *R_merge_* value is quite high, but the precision-indicating merging R(factor) values (i.e., the *R_pim_*) are quite reasonable, as the *R_pim_* describes the precision of the averaged measurement (see Weiss, (2001)). The CC_1/2_ value for the highest resolution shell is above 0.50, a generally accepted cutoff value for high resolution data shells.

R_merge_ should be reported in percent (not 0.085 but 8.5% ). R_work_ and R_free_ should only be reported for the entire dataset, omit highest resolution shell.

This has been fixed, see the revised Figure 1—source data 1. As the PDB requests these statistics for the overall dataset and the highest resolution shell during the deposition process and records these values in the header of the PDB deposition, we are following the PDB standard format and reporting those statistics in Figure 1—source data 1.

As the structure does not contain ligands or ions, the corresponding atomic and B-factor values can be omitted. 356 water molecules modeled into a structure at 2.7 Å resolution appears unusually high, especially with weak or no data in the highest resolution shell. Were waters modeled into noise? Limit the placement of water molecules to a sensible B-factor.

The ratio of total number of protein residues/water molecules is 2,275/356 = 6.4. A ratio of one water molecule to approximately 6 amino acids is not totally unreasonable for a 2.70 Å map, especially given that the mean refined B-factor for the waters is 36.1 Å^2^ (and the maximum value is ~58 Å^2^, which corresponds well with the mean refined B-factors for the Imp9 molecule). The |2F_o_ – F_c_| electron density map, contoured at the 1 r.m.s. level, was visually inspected to ensure that all modeled waters displayed significant electron density associated with each water, and that the modeled waters made appropriate hydrogen bond donor/acceptor interactions with the protein. These modeled waters are primarily in the first hydration shell of the Imp9 protein (chains A and D), which is the best-ordered portion of the model (as evidenced by the lower B-factor values).

The Molprobity and Ramachandran statistics are missing, these are essential quality control values that must be reported in the table. For both datasets the number of total and unique reflections are missing.

We added the requested, see revised Figure 1—source data 1.

Finally, the PDB code is not reported, the authors should formally submit the structure to the PDB and include the accession number in the revised manuscript.

This has been added, see revised Figure 1—source data 1. The PDB ID is 6N1Z.

Within the methods portion for the crystallization and structure determination both the number of molecules in the ASU and how NCS has been treated during refinement should be detailed.

This has been added, see revised Materials and methods section.

Reviewer #2:[…] 1) The authors make a strong point in their Discussion section that the histone tails would not be recognized by importin-9, based on the fact that they don't see an electron density for the interaction. Nevertheless, the binding strength drops by substantial factors when either the H2A or the H2B tail is deleted (4.5-fold and 13-fold, respectively). The combined effect of both deletions will cause a 60-fold drop in binding strength. This is certainly not a minor effect. Given that the authors measured the affinities at a higher than physiological ionic strength, the drop will be even greater under import conditions. Dismissing the tails' contribution is therefore a misinterpretation.

All three reviewers had substantial concerns about our previous ITC data. We took their concerns seriously and repeated all ITC experiments using single batches of each of the different proteins to decrease variations that might arise from different preparations of protein reagents. Every set of the interactions was evaluated in triplicate. The new data are summarized in the new Table 1 and the thermograms, traces and fits for the triplicate experiments are shown in Figure 1—figure supplement 1.

The new ITC experiments were performed much more rigorously with much higher quality proteins (gels, gel filtration traces and mass spectrometry data shown in this response document) in buffers with 150 mM NaCl, and the data are consistently cleaner. We now see that removal of the H2A tail (H2AΔTail-H2B) or H2B tail (H2A-H2BΔ(1-35)) or removal of both N-terminal tails (H2AΔTail-H2BΔTail) did not affect binding affinity for Imp9. Careful examination of our further refined final structure revealed weak electron density and very high B-factors (>100 Å^2^) for the short segment of the H2B tail (residues 28-32) that contacts Imp9 at Interface 2 (described in the Results section and shown in Figure 2—figure supplement 2B and 2C). This structural observation is consistent with the mutagenesis/ITC results showing that neither H2A nor H2B tail contributes much to binding Imp9.

2) The authors should provide explanations for the missing electron densities. Obvious ones are: (A) that the precipitant used for crystallization is of very high ionic strength (3M potassium formate) and therefore likely to artificially break otherwise important salt bridges and (B) that the tails might engage in a fuzzy interaction with the importin. This is an established concept describing interactions of disordered regions with a globular protein. In the specific case, there are probably several possibilities for how the basic tails salt-bridge to negatively charged regions of the importin.

Our new ITC data show internal consistency. The data is also consistent with the absent electron density for most of the H2A and H2B N-terminal tails and the very weak electron densities (and very high B-factors) of residues 28-32 of the H2B tail. The H2A and H2B tails do not contribute much binding energy to interactions with Imp9 in solution at physiological ionic strength (ITC performed in buffer with 150 mM NaCl).

It is formally possible that the very high ionic strength of the crystallization condition may artificially prevent dynamic/fuzzy electrostatic interactions between Imp9 and histones in the crystal. But, this is not consistent with our mutagenesis and ITC results. That being said, we did detect very weak and dynamic interactions between an Importin and cargo by NMR that could not be observed by X-ray crystallography or detected in mutagenesis/ITC experiments (Kapβ2 and cargo FUS; Yoshizawa et al., 2018). Therefore, we added discussion(Discussion section) suggesting that very weak dynamic/fuzzy long-range electrostatic interactions between Imp9 and histones tails cannot be completely discounted based on the absence of electron density and the lack of effects observed in mutagenesis/ITC studies.

3) In the light of the just made points, I regard the comments on the Muhlhausser, (2001) paper (Discussion section) as inappropriate.

We assume the reviewer was referring to the first sentence in the previous Discussion section, which referenced the Mosammaparast et al., (2001) paper and not the Muhlhausser (2001) paper. We made extensive revisions to the first paragraph of Discussion section and no longer refer to either the Muhlhausser et al., or the Mosammaparast et al., papers.

That new paragraph now reads:

“The solenoid-shaped Imp9 wraps around the folded globular domain of the H2A-H2B dimer, leaving most of the N-terminal tails of H2A, H2B and the C-terminal tail of H2A disordered in the complex. […] Nevertheless, the H2A-H2B dimer thus belongs to a small category of nuclear import cargos that use surfaces of folded domains rather than extended linear nuclear import/localization motifs to bind their importins (Aksu et al., 2016; Bono et al., 2010; Cook et al., 2009; Grunwald and Bono, 2011; Grunwald et al., 2013; Matsuura and Stewart, 2004; Okada et al., 2009).”

4) Likewise, the authors give the impression as if they had contradicted the concept of importins functioning as chaperones for highly basic cargoes (Discussion section). The concept (Jäkel et al., 2002) implied that importins shield histones and ribosomal proteins against aggregation with polyanions and gave an intuitive explanation as to why import signals are not always as short as an SV40 NLS. It is unclear why the authors phrase their data as a contradiction. It would be more appropriate to give proper credit; the importin-chaperone concept should already be introduced in the Introduction.

We did not intend to contradict the concept of Importins functioning as chaperones for highly basic cargos. Instead, we want to expand the concept of chaperoning beyond the obvious charged-charged interactions since we observe many hydrogen bonds and hydrophobic interactions (many more than electrostatic contacts) in our Imp9•H2A-H2B complex. We revised the Discussion section to emphasize that we are not contradicting but supporting the previously reported Importin-chaperone concept by Jakel et al.

That paragraph now reads:

“Görlich and colleagues proposed in 2002 that negatively charged importins act as chaperones toward positively charged cargo proteins like histones (Jakel et al., 2002a). […] Kapβ2-FUS interactions are anchored through high affinity binding at the 26-residue PY-NLS linear motif of FUS that then enable weak, distributed and dynamic interactions with multiple mostly intrinsically disordered regions of FUS, to block formation of higher-order FUS assemblies and liquid-liquid phase separation (Yoshizawa, 2018).”

5) The referencing in the introduction is inappropriately biased. For example, there is no mentioning of the first discovered import pathway for a histone (Jakel, 1999). Likewise, the by now established fact that the RanGTPase system determines the direction of transport is referenced by two reviews of the Chook lab – despite the fact that the RanGTPase gradient concept had been worked out by a set of papers by the Görlich lab (see e.g. Izaurralde el al., 1997). Please amend this and similar problems.

We added the two references suggested by the reviewer to the revised Introduction.

6) I have little faith in the model of the RanGTP/ importin-9 complex. The Chook lab published a similar prediction before, namely a model of how RanGTP binds to the CRM1/ snurportin 1 complex (Dong et al., 2009). The experimental structure of the RanGTP/CRM1/ snurportin complex (Monecke et al., 2009), however, revealed that the model was not correct. I cannot see why a prediction of the RanGTP/ importin-9 prediction would be any better. There is less than 20% sequence identity to the most similar importin/ exportin with a solved RanGTP structure. Furthermore, the mode of RanGTP-binding is only partially conserved amongst the members of the importin β superfamily. Some conservation is evident in the contact regions of HEAT repeats 1-3. The additional contacts, however, are not conserved; they differ widely between the various solved RanGTP complexes, and I cannot see how one could possibly predict new ones in any reliable manner. A validation of the modeled interaction by mutagenesis will also not help because such validation will not assess the most problematic aspects of the model, namely contacts that have not been predicted. The RanGTP/ importin-9 structure model should be omitted because there is no reason to believe that it is reliable.

As the reviewer indicated, the N-terminal repeats (HEATs 1-4) of karyopherins, especially amongst importins, are conserved (we show this in a sequence alignment in Figure 4—figure supplement 4G). HEAT repeats 1-4 of the 7 importins shown in Figure 4—figure supplement 3A-F are of much more conserved (mean sequence identity of 20%) than the full-length importins (mean sequence identity of 17%). The locations of RanGTP binding, with its switch 1, switch 2 and helix 3 contacting importin HEATs 1-4, are clearly similar for Impβ, Kap95, Kapβ2, Kap121, Transportin-SR2 and Importin-13 (Figure 4—figure supplement 3A-F). Ran also binds exportins at equivalent sites on their HEATs 1-4 (data not shown). Examination of details of the Ran binding sites in Figure 4—figure supplement 4A-E and the sequence alignment Figure 4—figure supplement 4G show structural conservation of Ran binding at the N-termini of the importins.

In Figure 4—figure supplement 3G we show only the N-terminal HEAT repeats of Imp9, with H2A-H2B bound and Kap121•Ran aligned to mark the predicted Ran site, which looks just like the equivalent sites in panels A-F. This clearly marked “model” is shown to suggest that the Ran site on Imp9•H2A-H2B is most likely accessible. We do not show the bottom half of the Imp9•H2A-H2B structure so as not to imply how that half of Imp9 might bind RanGTP; we do not know the extent of how the superhelix of Imp9 might rearrange upon binding RanGTP. In Figure 4—figure supplement 3A-F, we observe that the superhelices/solenoids of the Importins adopt different pitches thereby placing different parts of the C-terminal halves of these Importins to interact with the α-4 helix and basic patch of RanGTP.

(Note: RanGTP was modeled onto the N-terminal HEAT repeats of the CRM1•SNUPN complex (Dong et al. NSMB 2009). The location of the predicted site at HEATs 1-4 of CRM1 resembles that in the Ficner/Görlich structure but the overall structure of our RanGTP•CRM1•SNUPN model looks different because of the large conformational change in the CRM1 superhelix that placed its C-terminal HEAT repeats to clamp RanGTP on the opposite side).

7) The signal to noise ratio in the ITC experiments is rather poor – resulting in rather broad confidence intervals of the fits. Some plots e.g. Figure 1—figure supplement 1D, E, G, H failed to reach saturation. Please supply a substantially improved dataset. Since the impact of ionic interactions and the magnitude of the effects are important for the argument, a dataset at physiological salt should be included. For measuring weak interactions, it will help to raise the concentration of the titrant to 30-50-fold higher than the analyte.

We took the reviewers’ concerns very seriously and repeated all ITC experiments using single batches of the different proteins to decrease variations that may arise from different preparations of protein reagents. The new ITC experiments were performed more rigorously with much higher quality proteins (gels, gel filtration traces and mass spectrometry data shown in this response document) in buffers with 150 mM NaCl. Every set of interactions was evaluated in triplicate and the data are consistently cleaner. The new data are summarized in the new Table 1 and the thermograms, traces and fits for the triplicate experiments shown in Figure 1—figure supplement 1.

Some interactions just do not result in high ΔH. Fortunately, none of our interactions are weak enough to warrant drastically raising titrant concentrations so we could avoid problems like precipitation and poor c-values that come from very high protein concentrations. Bad heats of dilution for some samples are something that plagues the entire histone field, and we have done what we can to address it.

8) Please identify the used H2A and H2B variants by including the relevant UniProt numbers.

After further consideration, we feel that the section on histones variants really does not fit in the current story. We removed this entire Results section. We will pursue more rigorous biochemical and cellular studies of Imp9 interactions and nuclear import of H2A and H2B variants for a future report.

9) Figure 1A (upper panel). Please identify the protein bands.

We re-did the experiment in Figure 1A and labeled the protein bands in new Figure 1A.

10) Figure 1B, please improve clarity of the cartoon.

We moved Figure 1B to the supplement (reviewer #1’s suggestion) and improved clarity of the cartoon by making it larger.It is now the new Figure 1—figure supplement 1A.

11) Figure 1E-G are too crowded to get the information. Please re-label to improve clarity, and please label only what is really needed for argument.

To improve clarity of the old Figures 1E-G (now new Figures 2B-D), we

1) Enlarged each figure panel.

2) Removed labels for many side chains that are partially hidden, thereby relieving the crowding.

We also added stereo figures of broader views for each of the three interfaces in the supplement. These new figures are now in Figure 2—figure supplement 1A-C.

We also added images showing omit map electron density of loops (omitted to calculate the omit maps) in each of the three interfaces. These new figures are now in Figure 2—figure supplement 2A-D.

12) Please clarify which form of Ran (wt vs. Q69L, yeast of human) is used for which experiment.

The information for the Ran used was already detailed in the Methods section of the original manuscript. We used *S. cerevisiae* Ran (Gsp1) residues 1-176 with the Q71L mutation. The truncated protein is stabilized in the GTP state and the Q71L mutation abolishes GTPase activity further stabilizing the GTP state. We added this information to the main text (Results section) and to the Figure 4 legend.

13) Fitting of the models into SAXS data is not very conclusive. How well do alternate coordinate models fit the SAXS data?

We removed the SAXS model from the new Figure 4. It is unclear how we should make alternate models for the Ran complex. But, independent of the model, the MW weights of the complexes determined by SAXS indicate that RanGTP is indeed bound to Imp9•H2A-H2B – we kept this data (upper portion of old Figure 3E) as the new Figure 4F.

14) The term "NLS" is ambiguous and by many (if not most) cell biologists considered to be an importin α-dependent nuclear import signal. I would therefore refrain from calling the importin 9-interacting interface an NLS. The more generic term "import signal" should cause less confusion.

We replaced “NLS” in the first paragraph of Discussion section with “extended linear nuclear import/localization motifs”.

Reviewer #3:[…] Imp9 does not bind the NLS like sequences on the histone tail, but covers its surface. This deserves a a bit more discussion and a more rigorous analysis, including in ITC assays, as will be suggested below.

We added materials in the text on Imp9 covering the surface of the histone core rather than binding NLS-like tail sequences, in the following manner:

1) We expanded the description of the structures of the Imp9•H2A-H2B interfaces in Results section. We show and describe the weak electron density and very high B-factors of the small portion (5-residue segment) of the H2B tail that contacts Imp9 in Interface 2. We gave more details about Interfaces 1 and 3, including a new figure supplement (Figure 2—figure supplement 1D,) that shows many contacts made by the main chain of Imp9 to the histones.

2) We also show very consistent new mutagenesis/ITC data (Table 1 and Figure 1—figure supplement 1), which clearly show that the N-terminal tails of H2A and H2B tails do not contribute significantly to Imp9 binding. These data are described in the Results section.

3) We expanded discussion in the first and third paragraphs of Discussion section.

Additional experiments to test the existence of the Imp9-H2A-H2B complex in both cytoplasm and nucleus would strengthen the reported interaction at the level of IP in HeLa cell extracts.Essential revisions:ITC data:- The results appear to have been derived from single measurements, see Figure 1—figure supplement 1A. As can be seen from this example, there are anomalies in data points in the important region of the titration where the heat of injection changes most. This indicates that the data point for full-length H2A-H2B cannot be trusted. Figure 1—figure supplement 1D, E, and F suffer from similar problems. I would urge the authors to conduct these assays in triplicate. In the absence of these data, the comparison between the distinct H2A-H2B constructs appears premature. I encourage the authors also to provide a stoichiometry value, as well as the ΔH and ΔS parameters from the fitting of the results.

We took the reviewers’ concerns very seriously and repeated all ITC experiments using single batches of each of the different proteins to decrease variations that may arise from different preparations of protein reagents. The new ITC experiments were performed much more rigorously with much higher quality proteins (gels, gel filtration traces and mass spectrometry data shown in this response document) in buffers with 150 mM NaCl. Every set of interactions was evaluated in triplicate experiments and the new data are consistently cleaner. The new data are summarized in the new Table 1 and the thermograms, traces and fits for the triplicate experiments are shown in Figure 1 —figure supplement 1.

In the SEDPHAT analysis, we fixed the stoichiomnetry in a 1:1 model. Therefore, instead of reporting stoichiometry, we report the Imp9 concentration-correction errors. We have provided ΔG, ΔS, and ΔH values in the new Table 1.

- The ΔH18-19 loop deletion construct shows a quite distinct ITC profile, the fitting here was carried out using an endothermic reaction, rather than fitting exotherms. This is not discussed in the manuscript, but ought to briefly be explained.

We made a brief mention of the endothermic vs. exothermic reaction of the ΔH18-19 loop deletion construct in Results section.

- Point mutants might also be quite instructive compared to the rather less surgical deletion of entire histone tails and loops, even if their impact to the overall binding energy may be limited.

Because of the extensive (and three separate) binding interfaces and the presence of many interactions in all three interfaces that involve main chain atoms, we agree with the reviewer that the impact of point mutants to overall binding energy will be limited. Because of the questionable utility of making Imp9 point mutants, we focused our energy on re-doing all the ITC analysis of deletion mutants of Imp9 and histones in the most rigorous manner.

(Note: considering the aforementioned limitations on the accuracy of the ITC experiments, providing four significant figures appears unjustified)

We decreased the significant figures of ITC parameters in the new Table 1.

Pulldown experiment between importin 9 and H2A-H2B constructs:- Figure 2D lacks an appropriate negative control, such as an unrelated protein, MBP alone and/or an important mutant/distinct isoform that does not bind H2A-H2B.

Controls were added to the new Figure 4—figure supplement 1. We added a gel (Figure 4—figure supplement 1C) that shows controls of Imp9, H2A-H2B and RanGTP added to MBP immobilized on amylose resin. None of these proteins bind non-specifically to MBP or to the amylose resin. This includes H2A-H2B, which does not bind to immobilized MBP (lanes 6-10, Figure 4—figure supplement 1C).

For the gels in Figures 4A and 4B, we show the inputs and also the flow-through (FT) materials of all the pull-down experiments in Figure 4—figure supplement 1A, B. Both the inputs and FTs show the large excess and also the increasing concentrations of RanGTP used.

-The authors claim that the Imp9-H2A-H2B complex is present in the cytoplasm and nucleus and possibly also on chromatin. However, they do not directly show this. Co-IPs in these different cellular fractions would confirm this and would be a nice addition to the paper.

Co-IP experiments in the old Figure 1A (using whole cell lysates) were repeated with cytoplasmic and nuclear fractions. The new data is shown in the new Figure 1A. Imp9 is detected almost exclusively in the cytoplasm fraction where it co-IPs with H2B^mCherry^. Only a trace of Imp9 is detected in the nuclear fraction and none is detected in co-IP with nuclear H2B^mCherry^. The co-IP results are consistent fluorescence microscopy imaging in Figure 1B showing that Imp9 is mostly localized to the cytoplasm.

*- Additional assays/evidence, such as imaging, showing that RanGTP can bind with Imp9-H2A/H2B* in vivo *would strengthen the manuscript.*

We performed co-IP with H2B^mCherry^ of the nuclear fraction. Ran is abundant in the nucleus and co-IPs with H2B^mCherry^. However, Imp9 is hardly detectable in the nuclear fraction even though it is abundant in the cytoplasmic fraction. Imp9 co-IP with H2B^mCherry^ is detected only in the cytoplasmic fraction. Similarly, imaging also shows that Imp9 is found mostly in the cytoplasm. Imp9 is likely in the nucleus only transiently.

Gel shift assay in Figure 3B:- As it stands, the current assay is difficult to interpret, as samples are not shown comparable on the same gel. In addition, the conditions that contain Imp9 are different, one is a preformed complex and the other is not. This also makes it difficult to compare activity and make firm conclusions based on this preliminary experiment. This can be readily experimentally addressed.

We performed new and extensive gel shift assays, showing all samples from an experiment on the same gel. These new results are shown in:

1) Figure 3D and E – Imp9 and DNA competing for H2A-H2B (ethidium bromide and Coomassie gels, respectively).

2) Figure 4D – Imp9 titrated to H2A-H2B to show formation of the Imp9•H2A-H2B complex and Ran to Imp9 or Imp9•H2A-H2B to show Imp9-Ran interactions and formation of the heterotetrameric Ran•Imp9•H2A-H2B complex.

3) Figure 5A and 5B – DNA is titrated to Imp9•H2A-H2B or RanGTP•Imp9•H2A-H2B, showing that DNA cannot compete for H2A-H2B from Imp9•H2A-H2B but can compete for H2A-H2B from RanGTP•Imp9•H2A-H2B to result in Imp9•RanGTP. Coomassie staining in 5A, and ethidium bromide staining in 5B.

4) Figure 5C and 5D – Imp9 or Imp9•Ran is titrated to DNA•H2A-H2B. Imp9 releases much more DNA than Imp9•Ran. Ethidium bromide staining in 5C, Coomassie staining in 5D.

5) Figure 5E – DNA/ethidium bromide gel of nucleosome assembly assays (protein gel in Figure 5—figure supplement 1B; controls in Figure 5—figure supplement 1A).

6) Figure 5F – DNA/ethidium bromide of nucleosome disassembly assays (protein gel in Figure 5—figure supplement 1C; controls in Figure 5—figure supplement 1A).

Figure 4—figure supplement 1:- There seems to be less H2A/H2B when RanGTP is added. This experiment should be shown on one gel to compare conditions properly. In addition, this referee would suggest adding more RanGTP to test whether H2A-H2B levels decrease further.

We repeated the experiment by titrating increasing RanGTP concentrations, starting with 5 molar equivalents of RanGTP to MBP-Imp9•H2A-H2B in lane 6 of Figure 4A followed by 10 molar equivalents in lane 7, 20 molar equivalents in lane 8 and 30 molar equivalents in lane 9. Each of the four lanes show similar amounts of H2A-H2B and RanGTP bound to immobilized Imp9. Little H2A-H2B is displaced from the beads in the FT as shown in lanes 24-27 of the gel shown in Figure 4—figure supplement 1A.

The same amount of excess RanGTP was used in the experiment of MBP-PY-NLS binding to GST-Kapβ2 (Figure 4—figure supplement 1B). In this case, even 5 molar equivalents of RanGTP easily released all previously bound MBP-PY-NLS from the immobilized GST-Kapβ2.

[Editors' note: the author responses to the re-review follow.]

[…] We judge that there is not yet sufficient evidence that the RanGTP:Imp9:H2A:H2B complex is stable in solution, raising the question whether the proposed mechanism is correct that RanGTP binding to the Imp9:H2A:H2B complex is required for priming of the import complex that it can serve as a H2A:H2B donor during nucleosome assembly. Therefore, my criticisms of the revised manuscript refer primarily to the insufficient quality of newly included size-exclusion chromatography (SEC) interaction analysis that was introduced to support the formation of a stable RanGTP:Imp9:H2A:H2B tetramer.Figure 4—figure supplement 1D. We had requested that the authors carry out additional SEC interaction experiments, ideally combined with a multi-angle light scattering analysis, that would clearly establish that RanGTP binding indeed results in the formation of a heterotetrameric RanGTP:Imp9:H2A:H2B complex, rather than triggering the release of the bound H2A:H2B import cargo and the formation of a heterodimeric Imp9:RanGTP complex, as it has been observed in countless other canonical import complex cases. This analysis is at the heart of the novelty of the presented manuscript. While the authors have included such a SEC analysis, the results of the experiment are uninterpretable. Either the quality of the experiment is insufficient, or the experiment is contrary to the author's conclusion.The two SDS-PAGE gels of the preincubation run fractions of a mixture of Imp9:H2A:H2B with a 4-fold molar excess of RanGTP do not show stochiometric incorporation of RanGTP into the Imp9:H2A:H2B complex. As judged by the gel filtration profile of RanGTP in isolation, RanGTP elutes as two peaks, consistent with its known weak dimerization behavior: a major peak at 18.2 ml and a second much smaller peak at ~14 ml. The second smaller peak is close to the position where the Imp9:H2A:H2B heterotrimer elutes (12.9 ml). While there is a faint RanGTP band in the SDS-PAGE gel of the preincubation run (lanes 13-16), the band is clearly much weaker than the H2A:H2B bands, indicating at best a sub-stochiometric incorporation of RanGTP into the Imp9:H2A:H2B complex. However, it seems that the faint RanGTP band may not be the result of the incorporation of RanGTP into the Imp9:H2A:H2B complex, but is rather the secondary peak of RanGTP co-eluting independently. An SDS-PAGE gel of the RanGTP control injection is essential to distinguish these two possibilities. Moreover, there is a shift of ~0.4 ml in the elution profiles of RanGTP in isolation compared to the preincubation run profile (this is best seen in the blue and brown major peaks of RanGTP at an elution volume of ~18 ml). Such differences are common when experiments are carried out on different gel filtration columns, with different tubing lengths, or on different FPLC instruments. Additionally, the amounts of the injected proteins are unequal in the various injections and the baselines indicate that the column was not properly equilibrated between different injections. To definitively conclude whether RanGTP stoichiometrically incorporates into the Imp9:H2A:H2B complex, a proper SEC analysis needs to be carried out in which the various components are injected subsequently on the same gel filtration setup. High quality SDS-PAGE gels for all runs are essential for the interpretation of the experiment and should be included in the revised version of the manuscript.

At the request of the reviewer, we have performed new size exclusion chromatography (SEC) experiments to examine the interactions between Imp9•H2A-H2B and RanGTP. The SEC panel previously shown in Figure 4—figure supplement 1D of the old submission was an exploratory analysis performed very early in the project (in 2015), probably with somewhat subpar proteins. Our proteins preparations have improved tremendously since then. Here, we show two series of new SEC experiments in the new Figure 4—figure supplement 2.

The first SEC series include experiments (Figure 4—figure supplement 2A) with: (1) Individual proteins – 20 µM Imp9 alone and 60 µM RanGTP alone, (2) Imp9 + H2A-H2B 1:1 (20 µM), and (3) titrations of Imp9 + H2A-H2B 1:1 (20 µM) with Ran at 0.5 (10 µM), 1 (20 µM), 2 (40 µM) or 3 (60 µM) molar equivalents. Imp9 alone elutes at 13.6 mL, while the 1:1 Imp9•H2A-H2B complex elutes at 13.5 mL. When RanGTP is added, we see the formation of a 1:1:1 RanGTP•Imp9•H2A-H2B complex. Addition of an equimolar amount of RanGTP causes the Imp9•H2A-H2B peak to shift from 13.5 mL to 13.4 mL. Continued addition of RanGTP beyond a 1:1:1 mixture, results in the appearance of free RanGTP that elutes at 17.1 mL. Comparison to a 60 µM Ran alone control shows that the Imp9•H2A-H2B•RanGTP complex has a 1:1:1 stoichiometry. Quantitatively, the free RanGTP peak is absent in the 1:1:1 sample, is one-third of the control in a 1:1:2 sample, and two-thirds of the control in a 1:1:3 sample. The Ran only control shows that RanGTP is monomeric with no visible dimers. Free Ran elutes far from where Imp9 elutes.

The second SEC series is with proteins at higher concentration for SDS-PAGE visualization of eluted fractions (Figure 4—figure supplement 2B). We ran 1:1 Imp9 + H2A-H2B (70 µM) and 1:1:1 Imp9 + H2A-H2B + Ran (70 µM) on the same column with the same buffer as above, in two separate experiments. SEC of 1:1 Imp9 + H2A-H2B produces a single Imp9•H2A-H2B peak at elution volume of 13.5 mL. SEC of 1:1:1 Imp9 + H2A-H2B + Ran produces a single 1:1:1 RanGTP•Imp9•H2A-H2B peak eluting at 13.4 mL and SDS-PAGE of peak fractions shows the presence of Imp9, H2A-H2B, and Ran. Each protein stains in proportion to that seen in the input lane (note that the same molar equivalent of RanGTP in the input stains weaker than the histones), consistent with the formation of a 1:1:1 complex. Also, in the SEC there is no free H2AH2B or free Ran.

In conclusion, in the presence of Imp9, H2A-H2B and RanGTP, we see no free or released H2AH2B and we clearly see the formation of a 1:1:1 RanGTP•Imp9•H2A-H2B complex.

We also performed SEC analysis of the N-terminally truncated Imp9 mutant that was designed to not bind RanGTP, which we now use to replace the old pull-down binding assay that tested Ranbinding. Figure 4—figure supplement 6 panels C-E show SEC of the MBP-Imp9D1-144 mutant: C) + RanGTP, D) + H2A-H2B, and E) + H2A-H2B and RanGTP. SEC analysis shows that RanGTP does not interact with the Imp9 mutant. No interaction is seen at μM concentrations even when RanGTP is added at a 6-fold molar excess. This is obvious from the SDS-PAGE analysis of SEC fractions, showing that the Imp9 mutant and RanGTP do not comigrate (Figure 4—figure supplement 6C). The Imp9 mutant protein is functional as the interaction is maintained with H2A-H2B. This is consistent with the crystal structure showing that the region spanning HEAT repeats 1-3 of Imp9 (residues 1-144) is only a very small portion of the very large Imp9•H2A-H2B interface (Figure 4—figure supplement 6D). Not surprisingly, like Imp9 mutant alone in Figure 4—figure supplement 6C, the histone-bound Imp9 mutant also does not bind RanGTP when the GTPase is added at a molar excess (Figure 4—figure supplement 6E).

SEC details are described in the Materials and methods section.

Unfortunately, the poor quality of the newly included SEC analysis casts further doubts on the validity of the SAXS analysis of the RanGTP:Imp9:H2A:H2B complex. At a minimum the current SEC analysis seems to indicate that the RanGTP:Imp9:H2A:H2B heterotetramer is non-stochiometric at the tested Imp9:H2A:H2B concentration of 100 μM. The maximum reported concentration of the RanGTP:Imp9:H2A:H2B complex in the SAXS analysis was 30 μM. Because no stochiometric RanGTP:Imp9:H2A:H2B tetramer was observed in SEC analysis at an injected concentration of 100 μM, which is comparable to the SAXS analysis, if one considers that injected samples are typically ~4-fold diluted on a gel filtration column, it seems unlikely that the SAXS analysis was carried out with a monodisperse sample of the RanGTP:Imp9:H2A:H2B heterotetramer. Unfortunately, the methods provide no information how the authors reconstituted the RanGTP:Imp9:H2A:H2B tetramer for their SAXS analysis. Generally, monodispersity and the formation of a stable species is essential for a meaningful SAXS analysis. Therefore, SAXS experiments are typically carried out either directly following elution from a gel filtration column or better yet by directly coupling the gel filtration column to the SAXS cell.

Our new SEC analysis, performed with 20 µM or 70 µM protein concentrations in the input, show stable 1:1:1 RanGTP•Imp9•H2A-H2B complex. Per the reviewer’s assumption of ~1 in 4 dilution in the eluate peak of an SEC experiment, the 1:1:1 RanGTP•Imp9•H2A-H2B complex appears to be stable even down at 5 µM (Figure 4—figure supplement 2A). Our SAXS samples of Imp9, Imp9•H2A-H2B, Imp9•RanGTP, and RanGTP•Imp9•H2A-H2B varied in concentrations as high as 31-43 µM, which are all significantly more concentrated than in our SEC analysis.

Details of SAXS sample preparation:

We include a new paragraph in Materials and methods section with details of SAXS sample preparation. All SAXS samples were prepared by SEC to exchange the proteins into SAXS buffer and to remove excess proteins. We show the profiles of SEC runs (performed in July 2017) that produced the SAXS samples in Figure 1. SDS-PAGE/Coomassie-staining of the fractions were performed to ensure proper formation of complexes but the gels were not recorded or kept so regrettably cannot be shown. SEC to prepare the SAXS samples were also performed on a Superdex 200 column but not the same column used in the new Figure 4—figure supplement 2. The buffers are also different, as SEC preparations for SAXS were performed in SAXS buffer (20 mM HEPES pH 7.3, 110 mM potassium acetate, 2 mM magnesium acetate, 2 mM DTT, and 10% glycerol). Nevertheless, the trends of elution volumes are similar with the elution volumes of RanGTP•Imp9•H2A-H2B > Imp9 binary complex > Imp9 (see Figure 1A) with no peaks for free histones or free RanGTP in the second SEC for the preformed RanGTP•Imp9•H2A-H2B sample (Figure 1B).

After the SEC analysis, all our SAXS samples were further improved through multiple steps, right before the data collection at the SSRL SAXS beamline, as follows:

**Author response image 1. respfig1:** SEC for SAXS sample prepation. A. SEC profiles of 1) Imp9 alone (blue trace), 2) a previously purified Imp9•RanGTP complex (light blue trace) and 3) a previously purified Imp9•H2A-H2B + excess RanGTP (green; the peak at ~ 16 ml is excess RanGTP). Elution volume for each of the Imp9-containing peaks is listed. B. Fractions for the Imp9 containing peak in the SEC of Imp9•H2A-H2B + excess RanGTP (green trace in A) were pooled and subjected to a second round of SEC to produce the SAXS sample for the heterotetrameric complex.

First, the 10% glycerol in the SAXS buffer protects the protein samples from any potential radiation damage during X-ray exposure (Kuwamoto et al., 2004) (also see Figure 4—source data 1). Our earlier studies show that 5-20% (v/v) glycerol concentrations do not affect protein compaction (Yoshizawa et al., 2018).

Second, all protein and buffer solutions were filtered through 0.1 µm membranes (Millipore) to remove any aggregates, right before each measurement at the SAXS beamline (LoPiccolo et al., 2015).

Third, we collected SAXS profiles at multiple concentrations ranging from 0.5 to 5.0 mg/ml, then the final merged SAXS profiles were basically obtained by extrapolation to zero concentration, eliminating any concentration dependence affected by potential aggregation.

We achieved a high level of monodispersity, with no aggregation or degradation/dissociation, for our SAXS samples. This high level of monodispersity is independently determined from the SAXS analysis. Linear Guinier plots (rightmost panels, Figure 4—figure supplement 3A-D) for all 4 samples and consistency of molecular weight estimations (Figure 4G and Figure 4—source data 1, Figure 4—source data 2) report directly and independently on the monodispersity and the stability of the SAXS samples. In other words, SAXS analysis provides an internal readout on the monodispersity of the samples that are being analyzed, independent of data from sample preparation by SEC. SAXS is by far the more rigorous biophysical analysis and a quantitative one compared to SEC.

We explain further the relevant SAXS analysis (Guinier plots and MM determination) in detail here:

1) The linearity of a Guinier plot reports on monodispersity of the sample, in terms of the particle size as supported by basic SAXS theories. If there was a mixture of species of particles (e.g. any mixture of Ran•Imp9•histones, Imp9•histones, Imp9•Ran, Imp9 and Ran) in solution, a final SAXS profile would become a population-weighted combination from each component in the mixture. In other words, each component in the mixture would contribute to the Guinier plot with a different slope value (corresponding to its *R_g_* value), leading to a curvature or non-linearity in the Guinier plot. Consequently, the Guinier plot cannot be and is not linear when the sample contains a mixture of particles with different sizes. We did not observe such a curvature or non-linear pattern in any of the Guinier plots obtained from our SAXS data (rightmost panels, Figure 4 —figure supplement 3A-D). Our Guinier plots look very good and linear, indicating that all our SAXS samples were highly monodisperse, homogeneous and stable in solution.

2) The molecular weight estimated (MM or MW(SAXS) in Figure 4G) from the SAXS profiles of the RanGTP•Imp9•H2A-H2B sample is consistent with having 1 copy of each of the 4 polypeptide chains in RanGTP•Imp9•H2A-H2B, which further supports formation and stability of that complex. Our calculation of MM from the extrapolated scattering intensity at zero angle I(0) using SAXSMOW should have an uncertainty of <10% (Piiadov et al., (2018) and Fischer et al., (2010).

3) According to an article by the Svergun group, a worldwide leading group in SAXS, "one of the most important overall parameters, which can be derived from small-angle X-ray scattering (SAXS) experiments on macromolecular solutions is the molecular mass (MM) of the solute. In particular, for a monodisperse protein solution, MM of the solute is calculated from the extrapolated scattering intensity at zero angle I(0). Assessing MM by SAXS provides valuable information about the oligomeric state and absence of unspecific aggregation in solution. The value of MM can either be estimated by comparison with a protein standard with a known MM or by determining the absolute scattering intensity using, e.g., water scattering. In both cases, knowledge about the solute concentration and about the partial specific volume of the protein is required. […] One of the most common applications of SAXS is the determination of the oligomeric state of the biomolecule (e.g. a protein or a macromolecular complex) or monitoring of aggregation or degradation processes, which can be readily done by assessing the MM value." (Mylonas and Svergun, (2007)).

The observed difference of the major sedimentation coefficient peaks in the AUC analysis of the Imp9:H2A:H2B complex in the absence and presence of a 3-fold molar excess of RanGTP seems to indicate heterotetramer formation. However, AUC analyses are typically performed at much lower concentrations than gel filtration interaction analyses (the authors also provide here no detail in the methods about their employed concentrations). While the pull-down and AUC data seem to support the interpretation that RanGTP is indeed incorporated into the Imp9:H2A:H2B complex without releasing the H2A:H2B cargo, I am puzzled why the gel filtration data does not support this conclusion.

Individual purified proteins Imp9, H2A-H2B and RanGTP were dialyzed into AUC buffer before mixing them for AUC. Samples for AUC contain: 1) 450 µL Imp9 alone (3 µM), 2) 450 µL RanGTP alone (10 µM), 3) 450 µL H2A-H2B (10 µM), 4) 3 µM Imp9 + 3 µM RanGTP in a total volume of 450 µL, 5) 3 µM Imp9 + 3 µM H2A-H2B in a total volume of 450 µL, 6) 3 µM Imp9 + 3 µM H2AH2B+10 µM RanGTP in a total volume of 450 µL. The proteins were mixed overnight before loading into the AUC cell.

We have added these details of sample preparation to methods (pages 60-61). Proteins used for AUC are only slightly lower in concentration (3 µM) than from SEC analysis ((~5 µM). Both AUC and SEC analyses show RanGTP interacting with Imp9•H2A-H2B. AUC is by far the more rigorous biophysical analysis and a quantitative one compared to SEC. AUC shows the RanGTP•Imp9•H2A-H2B to be a larger assembly (4.6 S) than Imp9•H2A-H2B (4.3 S).

The recommendation is that the authors carry out a proper gel filtration interaction analysis as outlined above. As an important control, the SEC interaction analysis should also include the truncated Imp9:H2A:H2B complex that the authors identified to be deficient in RanGTP binding.

Figure 4—figure supplement 6 panels C-E show SEC of the MBP-Imp9D1-144 mutant in the presence of excess RanGTP, excess H2A-H2B, and of the purified MBP-Imp9D1-144•H2A-H2B + 3-fold molar excess RanGTP, all performed in buffer containing 20 mM HEPES pH 7.4, 200 mM sodium chloride, 2 mM magnesium acetate, 2 mM DTT and 10% glycerol. The MBP-Imp9D1-144 mutant does not bind RanGTP but is functional as it binds H2A-H2B. Consistently, MBP-Imp9D1144•H2A-H2B also shows no interactions with RanGTP.

Additionally, the entire SAXS analysis should be removed from the manuscript and the structural characterization of the RanGTP:Imp9:H2A:H2B complex and the likely associated conformational change upon RanGTP binding should be published separately in a future study. Alternatively, an EM analysis could be included but I do not think this is necessary.

We have performed extensive SEC analysis with many controls and at different protein concentrations, all of which show that 1) RanGTP does not dissociate H2A-H2B from Imp9 and 2) RanGTP binds the Imp9•H2A-H2B complex to form a 1:1:1 RanGTP•Imp9•H2A-H2B complex even in at an estimated concentration of ~5 µM. Compared to SEC, AUC analysis is a much more quantitative and rigorous method that shows the 3 µM RanGTP•Imp9•H2A-H2B complex to be larger than the Imp9•H2A-H2B of the Imp9•RanGTP complexes. SAXS samples were prepared by SEC and are of much higher concentrations (31-43 µM) than either AUC samples or the ~5 µM 1:1:1 stable RanGTP•Imp9•H2A-H2B complex that eluted from SEC. Most importantly, the ability of SAXS to accurately determine molecular weight of the macromolecular particle in a buffer with complex composition (e.g. buffers with glycerol that are critical for solubility of karyopherins) is superior to AUC and most certainly superior to SEC (method is subjected to problems such as shape of particle, interactions with column, viscosity/density of solution and macromolecular dissociation from shear force). SAXS analysis independently assesses mono/poly-dispersity of protein samples. Our SAXS analysis show that our samples, including that of the RanGTP•Imp9•H2A-H2B one, are highly monodisperse and the RanGTP•Imp9•H2A-H2B complex is stable with a MW(SAXS) or MM determined from SAXS profiles that matches a 1:1:1 assembly of Ran, Imp9 and H2A-H2B. SAXS analysis is in fact the most important complement to our AUC, SEC, EMSA and pull-down binding assays, and should not be removed from the manuscript.

We are unsure what aspect of "structural characterization of the RanGTP•Imp9•H2A-H2B complex” the reviewer is referring to. We have not solved or presented any structures of the RanGTP•Imp9•H2A-H2B complex. The closest thing to a “structure” in the previous submission was an *ab initio* SAXS model in a supplement figure. We have removed all *ab initio* SAXS models in Figure 4—figure supplement 3A-D. In the previous version of the manuscript, we also placed a cartoon/ribbon diagram of RanGTP into the N-terminal region of Imp9 through structural alignment of HEAT repeats 1-4 of Kap121 with Imp9. We have now removed the cartoon of RanGTP and replaced it with a schematic diagram to show the approximate placement of Ran from alignment with the Kap121•RanGTP structure (new Figure 4—figure supplement 6A). In a complementary figure we show only the structure of Imp9•H2A-2B that we solved and colored Imp9 residues at the predicted Ran-binding site green (new Figure 4—figure supplement 6B).

It is clear from DNA competition and nucleosome assembly/disassembly assays that RanGTP binding to Imp9•H2A-H2B changes the interactions between Imp9 and H2A-H2B. We therefore make a reasonable speculation that "the likely associated conformational change upon RanGTP binding” changes interactions between Imp9 and H2A-H2B. This speculation is limited to a single sentence in Discussion that states “Accessibility of the N-terminal HEAT repeats of Imp9 in the histones complex may allow formation of the RanGTP•Imp9•H2A-H2B complex, but proximity of the Ran and histones binding sites coupled with the flexibility of the HEAT repeats architecture of Imp9 and the propensity for conformational changes likely changed the kinetics of Imp9-histone binding.” We did not change this sentence. It would be intellectually lazy of us to not discuss or speculate possible ways Ran binding could influence Imp9•H2A-H2B interactions.